# Edge-based graph neural network for ranking critical road segments in a network

**Debasish Jana[1]⍟\*, Sven Malama[1]⍟, Sriram Narasimhan[1,2], Ertugrul Taciroglu[1]**

**1** Samueli Civil and Environmental Engineering, University of California Los Angeles, Los Angeles, California, United States of America, **2** Samueli Mechanical and Aerospace Engineering, University of California Los Angeles, Los Angeles, California, United States of America

⍟ These authors contributed equally to this work.
\* dj93@ucla.edu

**Data Availability Statement:** Data for the U.S. system is obtained from the Environmental Systems Research Institute, Inc. (ESRI) https://hub.arcgis.com/datasets/esri::usa-freeway-system/explore?layer=1. The network information of

## Abstract

Transportation networks play a crucial role in society by enabling the smooth movement of people and goods during regular times and acting as arteries for evacuations during catastrophes and natural disasters. Identifying the critical road segments in a large and complex network is essential for planners and emergency managers to enhance the network's efficiency, robustness, and resilience to such stressors. We propose a novel approach to rapidly identify critical and vital network components (road segments in a transportation network) for resilience improvement or post-disaster recovery. We pose the transportation network as a graph with roads as edges and intersections as nodes and deploy a Graph Neural Network (GNN) trained on a broad range of network parameter changes and disruption events to rank the importance of road segments. The trained GNN model can rapidly estimate the criticality rank of individual road segments in the modified network resulting from an interruption. We address two main limitations in the existing literature that can arise in capital planning or during emergencies: ranking a complete network after changes to components and addressing situations in post-disaster recovery sequencing where some critical segments cannot be recovered. Importantly, our approach overcomes the computational overhead associated with the repeated calculation of network performance metrics, which can limit its use in large networks. To highlight scenarios where our method can prove beneficial, we present examples of synthetic graphs and two real-world transportation networks. Through these examples, we show how our method can support planners and emergency managers in undertaking rapid decisions for planning infrastructure hardening measures in large networks or during emergencies, which otherwise would require repeated ranking calculations for the entire network.

## Introduction

### Motivation

Roads are essential components of a transportation network that facilitate the efficient movement of goods, services, and people. However, road networks increasingly face challenges due

Minnesota state road network is obtained from the network repository https://networkrepository.com The remaining four transportation network datasets (Aachen, Edinburgh, Luxembourg, and Santa Barbara Transportation Networks) are acquired from Pyrosm, a Python library for OpenStreetMap data https://doi.org/10.5281/zenodo.4279527. The edge weights are assigned from travel times for all these transportation network and are obtained through the Google Distance Matrix API https://developers.google.com/maps/documentation/distance-matrix/overview.

**Funding:** The research award (Award No. 20220480) was received by SN and ET from the funding agency Los Angeles Bureau of Engineering (https://engineering.lacity.gov/). The funders had no role in study design, data collection, and analysis, the decision to publish, or preparation of the manuscript.

**Competing interests:** The authors have declared that no competing interests exist.

to natural hazards such as floods or fires, which have become more frequent and severe due to climate change [1, 2], which has led to increased traffic disruptions, resource inefficiencies, and compromised safety. For example, after Hurricane Katrina in 2005, roads along the I-10/I-12 corridor remained impassible for many months after the event [3]; many services in New Orleans did not regain even half of their pre-Katrina capacity after two years [4]. Therefore, it is crucial to understand road network vulnerability and develop effective resource planning and risk management strategies to mitigate such monumental impacts.

Road network vulnerability analysis is a crucial process that involves assessing the susceptibility of a transportation infrastructure to various risks and potential disruptions. This analysis plays a pivotal role in enhancing the resilience of road networks, ensuring their ability to withstand and recover from unforeseen events such as natural disasters, accidents, or deliberate attacks. By systematically evaluating the vulnerabilities within a road network, transportation authorities can identify weak points, assess potential impact scenarios, and develop effective strategies to mitigate risks. The importance of road network vulnerability analysis lies in its ability to inform decision-makers about potential vulnerabilities, allowing for the implementation of targeted measures to enhance the overall reliability and functionality of the transportation system. This proactive approach not only helps minimize the impact of disruptions on traffic flow and public safety but also contributes to the sustainable development of resilient and adaptive transportation networks in the face of evolving challenges.

Assessing the vulnerability of road networks poses significant technical challenges. Traditional methods often rely on computationally expensive algorithms, which render acquiring vulnerability metrics challenging. This challenge becomes even more pronounced when interdiction methods are employed [5]. Interdiction involves intentionally removing network elements to evaluate the resulting impact on network flow. The computational burden associated with such analyses further complicates timely network vulnerability assessment. Moreover, the computational burden becomes even more pronounced when dealing with uncertainty and the need for repeated vulnerability simulations, such as in the case of uncertain events like earthquakes. This limitation hinders our ability to perform repeated scenario or post-event analyses to develop network mitigation strategies.

Surrogate models often enable fast approximation of network vulnerability and resilience [6, 7], which can be advantageous in many situations involving decision-making under uncertainty and repeated simulations. Previous studies have extensively utilized betweenness centrality, a fundamental metric in graph theory, for vulnerability and resilience analysis [8–10]. The research findings suggest that betweenness centrality provides valuable insights into the importance of individual roads, which can indicate network vulnerability. Transportation planners and policymakers can effectively allocate resources and implement targeted interventions to enhance network resilience by identifying roads with high betweenness centrality. Using betweenness centrality as a proxy for assessing network vulnerability offers a practical approach to prioritizing critical roads and strengthening transportation systems' overall robustness.

Building upon this relationship between the betweenness centrality measure and vulnerability, we propose a novel and efficient method for calculating edge betweenness centrality in weighted road networks. By leveraging the power of graph neural networks (GNNs), our approach allows for a fast and accurate approximation of edge betweenness centrality. This computational efficiency makes our method particularly suitable for large-scale road networks, where traditional techniques face significant computational challenges, especially when repeated simulations are necessary. By establishing the correlation between edge ranking and important vulnerability metrics, our framework demonstrates its effectiveness in approximating network vulnerability practically and efficiently. This contribution enables better decision-making in resilience and post-disaster recovery applications.

## Literature review

Significant literature exists related to network component ranking for post-disaster recovery, where an optimal sequencing (ranking) for repairing components is necessary to maximize the speed of network recovery or resiliency. Vugrin et al. [11] presented a bi-level optimization algorithm for optimal recovery sequencing in transportation networks. Gokalp et al. [12] proposed a bidirectional search heuristic strategy for post-disaster recovery sequencing of bridges. Bocchini and Frangopol [13] presented a model for recovery planning for a network of highway bridges damaged by an earthquake by maximizing resilience and minimizing the time required to return the network to a target functionality and within a restoration cost. Road repair prioritizing based on damage [14], average daily traffic volumes [15], or network robustness [16, 17] have been proposed. Many heuristic and meta-heuristic optimization techniques have been employed to minimize the effect of extreme events and develop restoration programs [11, 18–25]. However, these studies are limited to small networks due to the significant computational overhead of the optimization methods employed. Some studies have been performed on large networks [22, 25, 26]; however, these are limited to a small number of disruptions in the network occurring at any time. As large-scale events like Hurricane Katrina have shown, many disruptions could co-occur following a significant event—Katrina impacted 15,947 lane miles of highway in Alabama, 60,727 in Louisiana, and 28,889 in Mississippi, August 2005 [27].

Transportation network criticality metrics and network disruption analyses have been summarized in these review papers [5, 28, 29]. Most studies on pre-disaster component ranking employ a network performance metric such as importance or criticality [30–32], adaptability or resilience [33, 34], or vulnerability [35]. Disruptions in a transportation network are simulated by modifying link(s) in the network, then evaluating the network's performance in the modified condition. This process aims to understand and quantify the modified link's relative importance to other links in the network. The aforementioned studies use this procedure and performance metrics to estimate the importance rank of road segments. De Oliveira et al. [36] ranked the streets in Rio De Janeiro regarding vulnerability, network reliability, and traffic congestion. Esfeh et al. [37] proposed a combined data-driven and analytical ranking approach for roads in a network using a vulnerability index that considers the spatiotemporal impact of incidents in one link to the surrounding links. These studies rank the road's importance based on network topology and traffic-related parameters.

Graphs offer an intuitive approach to rank critical road segments for emergencies and routine traffic situations. Measures such as edge betweenness centrality (EBC) [38] and node betweenness centrality (NBC) [39] are commonly used for this purpose. EBC provides valuable insights into the flow of information within a graph [40]. High-importance edges, characterized by higher EBC values, are significant in maintaining the information flow. Disrupting an edge associated with a large EBC can significantly impact the overall information flow in the graph [41]. Betweenness centrality has been explored for transportation networks in [42], where a dynamic betweenness centrality framework was proposed to identify real-time congestion and vulnerable areas in transportation networks. To evaluate their suitability, Oldham et al. [43] tested various centrality measures for different network classes. Derrible [44] demonstrated how betweenness centrality can provide insights for designing infrastructure systems. Altaweel et al. [45] investigated the relationship between future population scaling and past node centrality in a network. Demšar et al. [46] found a strong correlation between streets with high betweenness and highways/main roads in the Helsinki Metropolitan area. The validity of critical components obtained from EBC has been verified using open-source macroscopic traffic simulation frameworks such as SUMO (Simulation of Urban MObility) and

OpenStreetMap (OSM) [47]. Wu et al. [48] expanded upon the conventional Brandes approach by incorporating origin-destination (O/D) flow and gravity models into betweenness centrality calculations. These studies highlight the practical relevance and effectiveness of betweenness centrality in transportation networks. They provide valuable insights into ranking critical road segments and their contributions to network flow, supporting decision-making in transportation planning and management. Despite the significant advantages of the betweenness centrality metric, computing it to rank road segments in a large network of hundreds or thousands of nodes and edges remains challenging [49].

Recent advancements in computing and the availability of large data sets have resulted in powerful machine learning and deep learning methods. Such learning-based methods have been used in transportation networks for travel time prediction [50], traffic forecasting [51], and missing traffic data imputation [52]. Mendonça et al. [53] proposed a simple neural network with graph embedding to estimate the approximate NBC. GNN is a deep learning architecture that leverages the graph structure and feature information to perform various tasks, including node/edge/graph classification [54, 55]. Maurya et al. [56, 57] proposed a GNN-based node ranking framework, and Fan et al. [58] utilized GNNs to determine high-importance nodes. Park et al. [59] used GNNs to estimate node importance in knowledge graphs. In the current literature, these GNN-based methods have been shown to work exclusively on node centrality ranking on unweighted directed and undirected graphs.

## Gaps and contributions

Based on the literature review, the following gaps are identified in existing works in the context of network resiliency and post-disaster recovery research. First, most of the literature focuses on ranking interrupted components; only a few address the full network component ranking after interruption. However, due to the computational overhead, such studies are limited to small and medium-sized networks only (20-100 nodes and 50-200 edges). Next, studies have been limited to identifying the ranking of critical segments of a network to be recovered, assuming that all identified segments can be recovered; however, cases where some of the most critical segments cannot be recovered are not addressed. Hence, applying these techniques for large transportation networks in rapid decision-making, such as improving the network's resiliency following a minor disruption or post-disaster recovery planning, is limited.

Our main contribution in this paper is a new learning method based on GNN to aid in the rapid importance ranking of street segments in a large network affected by minor and major interruptions. Specifically, through GNN we estimate the edge importance ranking to quantify network modifications due to interruptions. Such interruptions are modeled using edge weights, restructuring of nodes and edge formation, and node/edge failures. Conventionally, GNNs aggregate node features of the neighboring connected nodes in the graph. Such repetitive aggregation captures the overall structural and neighborhood information of the node of interest. Our proposed method modifies the conventional GNN architecture to work on edges instead of nodes. This modification to the original edge-adjacency matrix leads to a unique representation. As EBC considers both the network topology and traffic data (using travel time as a surrogate) as edge weights, the GNN is trained to output the EBC-based road importance edge ranking after training over multiple scenarios of simulated network interruption events. We develop an edge-adjacency matrix by modifying the original GNN architecture to work with edges (instead of nodes). To the best of our knowledge, this is the first application of GNN to estimate the importance ranking of the road segments in a large transportation network.

Conventional road importance ranking algorithms are limited to small and mid-size networks as ranking is computationally intensive for large networks. Once trained, the GNN model proposed in this paper is swift and scalable at the inference stage; hence, it can rapidly estimate the road importance rank where quick decision-making is critical. To demonstrate the model's generalizability, we show the proposed method's performance on three types of synthetic networks and two different sizes of real-world transportation networks: the US freeway network and the Minnesota transportation network. The trained GNN model can perform edge ranking for static and dynamic graphs considering directed and undirected cases. We demonstrate that GNN-based ranking results are comparable with conventional methods such as centrality and vulnerability-based ranking while achieving road ranking significantly faster for large graphs.

## Organization

The remainder of this paper is organized as follows. First, in the 'Preliminaries' section, we present the basics of graph theory, EBC, and edge feature representation. Next, the working principles of GNN and information propagation in the learning stage are described. Subsequently, in the 'Proposed GNN Framework' section, we present the new GNN-based framework, which forms the core of the contributions claimed in this paper. In the 'Results' section, we evaluate the performance of the proposed approach on synthetic graphs. Next, we demonstrate the performance of two real-world transportation networks. Afterward, two potential applications of the proposed framework are discussed in the 'Applications' section. Finally, the conclusions are presented.

## Preliminaries

This section introduces the concepts and terminologies necessary to follow the material presented in this paper. The basics of graph theory are explained along with an introduction to the edge adjacency matrix, which describes the spatial connection between edges in a graph. A brief introduction to EBC is also provided, followed by details of edge feature representation in the graph topology. Finally, the basic concepts of Graph Neural Networks (GNNs), including how the information of edges is exchanged and accumulated, are described. The original GNN algorithm [60] proposes the message passing on nodes. In contrast, the message passing is on edges here, and the GNN predicts edge importance rank through the edge adjacency matrix and the edge feature vectors.

### Basics of graph theory

A graph $\mathcal{G}$, is defined as $(\mathcal{V}, \mathcal{E})$—here $\mathcal{V}$ denotes the set of nodes or vertices of the graph and $\mathcal{E} \subseteq \mathcal{V} \times \mathcal{V}$ symbolizes the edges. The neighbor set of node $i \in \mathcal{V}$ is defined as $\mathcal{N}_i := \{j \in \mathcal{V} : (i, j) \in \mathcal{E}\}$. The graph edges are weighted by $w_{ij}$ which are associated with $(i, j)$ for $i, j \in \mathcal{V}$—here $w_{ij} > 0$ if $(i, j) \in \mathcal{E}$ and $w_{ij} = 0$ otherwise. The vertex adjacency matrix (also known as adjacency matrix) $\mathcal{A}^{\mathcal{V}} = [a_{ij}^v] \in \mathbb{R}^{|\mathcal{V}| \times |\mathcal{V}|}$ of $\mathcal{G} = (\mathcal{V}, \mathcal{E})$ is defined as [61]:

$$a_{ij}^v = \begin{cases} 0, & \text{if } i = j \text{ or there is NO edge present between } i \text{ and } j, \\ w_{ij}, & \text{if } i \neq j \text{ and there is one edge present between } i \text{ and } j. \end{cases} \tag{1}$$

Graphs used to model transportation networks are considered either bidirected or undirected, where for an undirected graph $\mathcal{G}$, $w_{ij} = w_{ji}$ $\forall (i, j) \in \mathcal{E}$ $(j, i) \in \mathcal{E}$ and for a bidirected graph $w_{ij} \neq w_{ji}$. Directed graphs are a special case of a bidirected graph obtained by setting one of the edge weights $w_{ij} \to \infty$. $|\cdot|$ refers to the cardinality or the number of elements in the set.

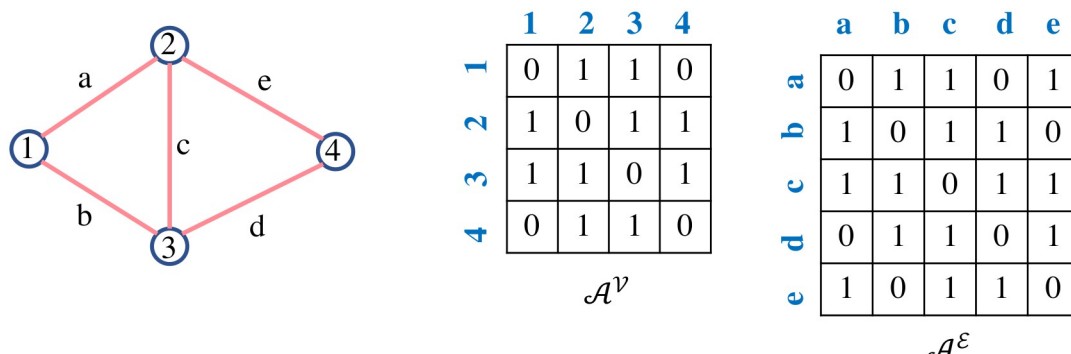

**Fig 1. Vertex adjacency matrix and edge adjacency matrix of a sample graph network.**

For unweighted graphs (for both bidirected or undirected cases), the weight values are 1, i.e., $w_{ij} = 1 \, \forall i, j$. Non-zero values sparsely appear in the vertex adjacency matrix when an edge exists between any two nodes. The unweighted edge adjacency matrix $\mathcal{A}^{\mathcal{E}} = [a_{mn}^e] \in \mathbb{R}^{|\mathcal{E}| \times |\mathcal{E}|}$ is determined by the adjacency of edges [62]:

$$a_{mn}^e = \begin{cases} 1, & \text{if edges } m \text{ and } n \text{ are adjacent}, \\ 0, & \text{otherwise}. \end{cases} \tag{2}$$

Fig 1 shows an example of the unweighted vertex adjacency matrix and the unweighted edge adjacency matrix for the same graph. It is to be noted that $\mathcal{A}^{\mathcal{V}}$ and $\mathcal{A}^{\mathcal{E}}$ are symmetric for both the undirected and bidirected unweighted graphs. The weights have been incorporated in the latter stage of the framework.

## Basics of EBC

Edge importance ranking depends on the edge's ability to control the information flow (the term information flow is contextual) between other nodes and edges of the graph and is highly correlated with the edge weights. The edge weights greatly influence the shortest paths calculated using the graph. Edge ranking based on this criterion is called EBC [40, 63]. The EBC score of an edge will be high if that edge contains many shortest paths making the information flow more accessible and faster throughout the whole graph. The edges with high betweenness centrality are called 'bridges'. Removing bridges from the graph can be disruptive; in some cases, one graph can segregate into several smaller isolated graphs.

For a given graph $\mathcal{G} = (\mathcal{V}, \mathcal{E})$, EBC of an edge $e$ is the sum of the fraction of all-pairs shortest paths that pass through $e$ and is given by [63]:

$$c_B(e) = \sum_{s,t \in \mathcal{V}} \frac{\sigma(s, t|e)}{\sigma(s, t)} \tag{3}$$

where, $\mathcal{V}$ and $\mathcal{E}$ are the set of nodes and edges, respectively, $s$ and $t$ are the source and terminal nodes while calculating the shortest paths. $\sigma(s, t)$ is the number of shortest $(s, t)$ paths and $\sigma(s, t|e)$ is the number of those paths passing through edge $e \in \mathcal{E}$.

The conventional method to calculate this betweenness centrality is through Brandes's algorithm [40]. This algorithm has a space complexity of $\mathcal{O}(|\mathcal{V}| + |\mathcal{E}|)$ and time complexity $\mathcal{O}(|\mathcal{V}||\mathcal{E}|)$ for unweighted networks. For weighted networks, the time complexity increases to

$\mathcal{O}(|\mathcal{V}||\mathcal{E}| + |\mathcal{V}|^2 \log |\mathcal{V}|)$ [64]. This algorithm is computationally prohibitive for large-scale networks (examples shown later in the 'Results' section). Additionally, this algorithm is sensitive to small perturbations in the network, such as changes in edge weights or regional node or edge failures. As a result, EBC is recalculated every time from scratch when there is a change in the graph. Hence, edge importance ranking estimation using EBC could be impractical for making rapid decision-making, e.g., in disaster response and recovery planning scenarios. To address this issue, in this paper, we pose the estimation of EBC as a learning problem and develop a deep learning-based framework whose time complexity is $\mathcal{O}(|\mathcal{V}|)$, hence can be utilized for large networks with perturbations.

## Node and edge feature embeddings

The adjacency matrices represent the connection information between the nodes and edges; however, the complete neighborhood information for nodes and edges is still incomplete beyond their immediate neighbors. The feature representation for nodes and edges embeds the knowledge of *k*-hop neighbors—hence the information is more exhaustive. Feature representation of the graph components is a way to represent the notion of similarity in graph components. Such embeddings capture the network's topology in a vector format which is crucial for numerical computations and learning. The most popular method for node embedding is `Node2Vec` [65].

`Node2vec` [65] is a graph embedding algorithm to transform a graph into a numerical representation. This algorithm generates a feature representation for each node that portrays the whole graph structure, such as node connectivity, weights of the edges, etc. Two similar types of nodes in the graph will have the same numerical representation in `Node2vec` algorithm. This representation is obtained through second-order biased random walks, and this process is executed in three stages:

1. *First order random walk*

   A random walk is a graph traversing procedure along the edges of the graph, best understood by imagining the movement of a walker. First-order random walks sample the nodes on the graph along the graph edges depending on the current state. In each step/hop, the walker transitions from the current state to the next referred to as a 1-hop transition. In Fig 2(a), the walker is at node *v* and three neighboring nodes are $u_1$, $u_2$, and $u_3$ with the respective edge weights, $w(v, u_1)$, $w(v, u_2)$, and $w(v, u_3)$. These weights determine the probability of the walker transitioning to the next node. The transition probability for the first step is given as,

   $$p(u_i|v) = \frac{w(u_i, v)}{\sum\limits_{u_i \in \mathcal{N}_v} w(u_i, v)} = \frac{w(u_i, v)}{\text{Degree of node } v} \qquad (4)$$

   Here, $\mathcal{N}_v$ is the set of neighboring nodes of *v*. One random walk is generated by performing multiple one-hop transitions; this process is repeated to multiple random walks, a function of the current state.

2. *Second-order biased walk*

   In the second-order biased walk, the edge weights selection differs from the first-order random walk. A new bias factor term $\alpha$ is introduced to reweigh the edges. The value of $\alpha$ depends on the current state, previous state, and potential future state, as shown in Fig 2(b). If the previous and future states are not connected, then $\alpha = \frac{1}{q}$, *q* is the in-out parameter. If

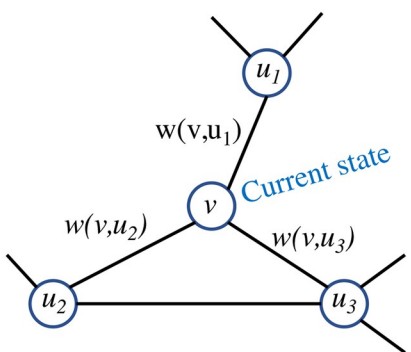 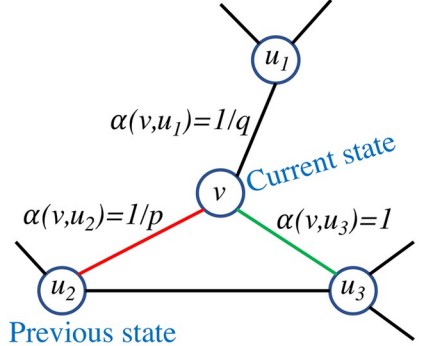 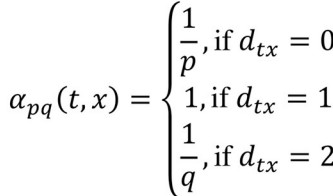

(a) First order random walk:
$v$ = current state
$u_i$ = neighboring nodes of $v$
(possible future state)
$w(v,u_i)$ = edge weights

(b) Second order biased walk:
$v$ = current state
$u_2$ = previous state
$u_i$ = neighboring nodes of $v$
(possible future state)

$$\alpha_{pq}(t, x) = \begin{cases} \dfrac{1}{p}, & \text{if } d_{tx} = 0 \\ 1, & \text{if } d_{tx} = 1 \\ \dfrac{1}{q}, & \text{if } d_{tx} = 2 \end{cases}$$

$\alpha$ = bias factor
$p, q$ = hyperparameters
$t$ = previous state
$x$ = possible future state
$d_{tx}$ = shortest path distance
between $t$ and $x$

**Fig 2. Conceptual representation of Node2Vec.** (a) Parameters for transition probability calculation for the 1st order random walk, and (b) parameters for transition probability calculation for the 2nd order biased walk.

the previous and future states are identical, then $\alpha = \dfrac{1}{p}$, where $p$ is the return parameter. If the two states (the previous state and the future state) are connected but not identical, then $\alpha = 1$. Considering the bias factors, the 2<sup>nd</sup> order transition probability is given as:

$$p(u_i|v, t) = \frac{\alpha_{pq}(t, u_i) w(u_i, v)}{\displaystyle\sum_{u_i \in \mathcal{N}_v} \alpha_{pq}(t, u_i) w(u_i, v)} \tag{5}$$

3. *Node embeddings from random walks:*
Repeated generation of random walks from every node in the graph results in a large corpus of node sequences. The `Word2Vec` [66] algorithm takes this large corpus as an input to generate the node embeddings. Specifically, `Node2vec` uses the skip-gram with negative sampling. The main idea of the skip-gram is to maximize the probability of predicting the correct context node given the center node. The skip-gram process for the node embedding is shown in Fig 3.
From the node embedding, the edge embedding is obtained using the average operator— edge embedder for $e(i, j)$ is $\dfrac{f(i) + f(j)}{2}$, where the edge ends are nodes $i$ and $j$; the node embedding of $i$ and $j$ are $f(i)$ and $f(j)$, respectively.

`Node2Vec` [65] can also be used for edge feature representation. The original implementation is found to be slow and memory inefficient [67]. Hence, a fast and memory efficient version of `Node2Vec` called `PecanPy` (**P**arallelized, memory **e**fficient and **a**ccelerated **n**ode2vec in **Py**thon) [67, 68] is utilized in this paper. `PecanPy` makes the `Node2Vec` implementation efficient on the following three fronts:

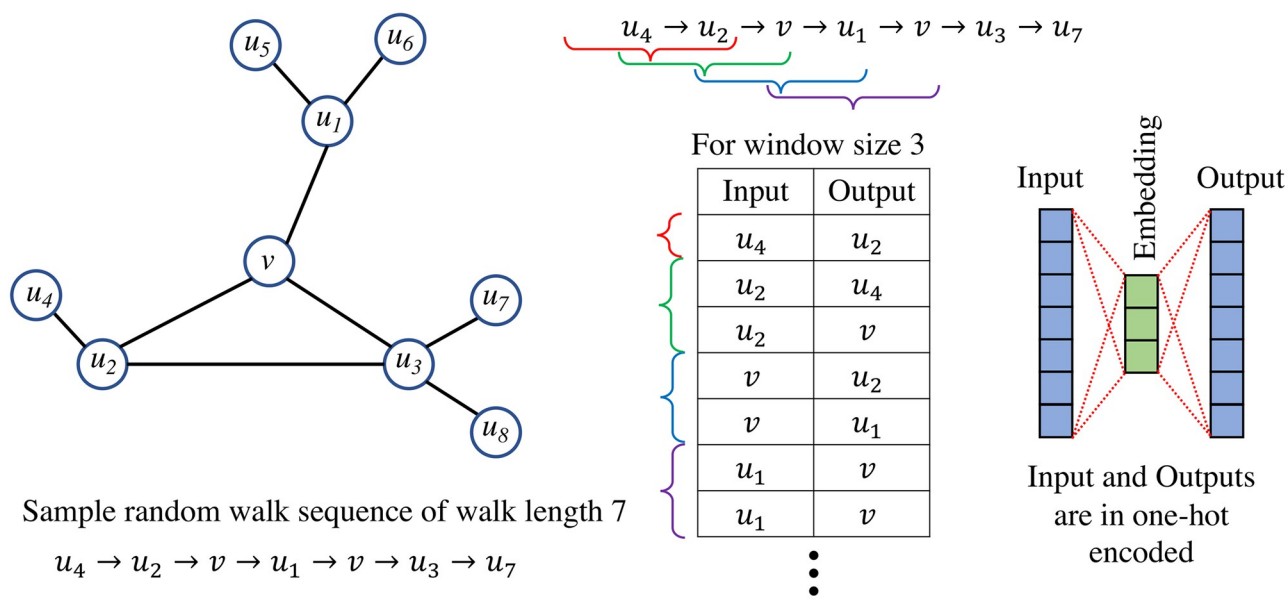

**Fig 3. This is an illustration of the skip-gram model for node embedding.** For a sample random walk of length 7, a sliding window of length 3 is used to prepare the inputs and outputs for training the `Word2Vec` model. The embedding of the trained `Word2Vec` model is the node feature embedding.

(a) **Parallelism**: The estimation of transition probability and the random walk generation processes are independent, but are not parallelized in the original `Node2Vec`. `PecanPy` parallelizes the walk generation process, which makes the operation much faster.

(b) **Data Structure**: The original implementation of `Node2Vec` uses NetworkX [69] to store graphs which is inefficient for large-scale computation. However, `PecanPy` uses the Compact Sparse Row (CSR) format for sparse graphs—which has similar sparsity properties to the transportation network that is addressed in this paper. The CSR formatted graphs are more compactly stored in memory and run faster as they can utilize cache more efficiently.

(c) **Memory**: The original version of `Node2Vec` pre-processes and stores the $2^{nd}$ order transition probabilities, which leads to significant memory usage. `PecanPy` eliminates the pre-processing stage and computes the transition probabilities whenever it is required, without saving.

`PecanPy` is capable of calculating the node and edge feature embeddings for both the undirected and bidirected graphs.

### Details of Graph Neural Network (GNN)

Neural network models for graph-structured data are known as GNNs [70]. These models exploit the graph's structure to aggregate the feature information/embeddings of the edges and nodes [71]. Feature aggregation from the structured pattern of the graph enables the GNN to predict the probability of edge existence or to predict node labels. The graph structure information is assimilated from the adjacency matrix and the feature information matrix of nodes and edges, which form the inputs, and training using a loss function. Message passing occurs in each GNN layer when each node aggregates the features of its neighbors. The node feature vector is updated by combining it with the aggregated features from its adjacent nodes. In the first layer, the GNN combines the features of its immediate neighbors, and with an increasing

number of layers, the depth of assimilating the neighboring edge features increases accordingly. The edge feature vector is updated with the aggregated features from its adjacent edges, and this procedure repeats for each GNN layer. There are three steps in a GNN, elaborated as follows:

- *Step 1: Message passing of edge features:*
  The original message-passing algorithm is applied to the node features passing via the graph's edges. Since this work focuses on ranking the edges, each edge is associated with edge features/embedding. The vector $\mathbb{R}^d$ represents such features as a latent dimensional representation. A popular algorithm to obtain such representation is `Node2vec` [65], as discussed previously. The new framework presented in this paper contains a modified version of the message-passing concept—edge features are aggregated and passed to the neighboring edges. In this way, the GNN learns the structural information. An example of the message passing step is shown in Fig 4. While conventional implementation of GNNs uses the message passing on the nodes, such passing is performed on the edges here, which is the novelty.

- *Step 2: Aggregation:*
  Messages are aggregated after all the messages from the adjacent edges are passed to the edge of interest. Some popular aggregation functions are:

$$\text{Sum} = \sum_{j \in \mathcal{N}_i} F(h_j); \quad \text{Mean} = \frac{\sum_{j \in \mathcal{N}_i} F(h_j)}{|N_i|}$$
$$\text{Max} = \underset{j \in \mathcal{N}_i}{\text{Max}}\, F(h_j); \quad \text{Min} = \underset{j \in \mathcal{N}_i}{\text{Min}}\, F(h_j) \tag{6}$$

where, $\mathcal{N}_i$ is the set of neighboring edges of edge $i$ (edge of interest). Considering

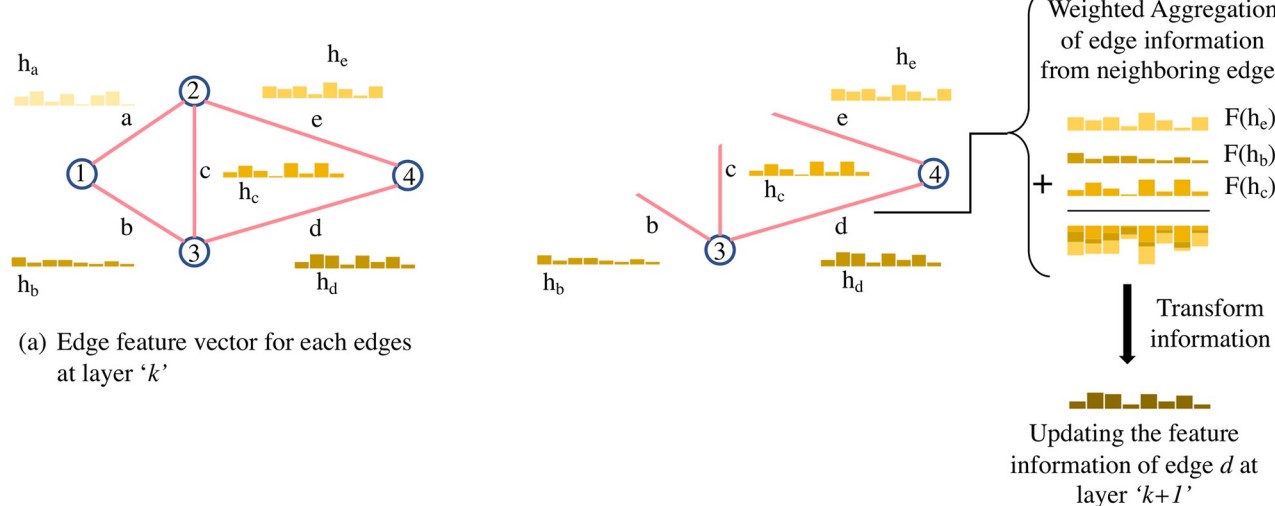

(a) Edge feature vector for each edges at layer '*k*'

(b) Edge feature updating in one GNN layer for edge *d*

**Fig 4. Schematic diagram of the message passing of GNN.** (a) This is a sample graph with 4 nodes and 5 edges. Each edge has its own embedding vector shown as $h_i$ for $i$th node, $i \in \{a, b, c, d, e\}$; (b) message passing procedure of edge $d$ using the edge vectors of the neighboring edges $b$, $c$, and $e$ and transforming them, finally "passing" them to the edge of interest. This process is repeated, in parallel, for all the edges in the graph. The transformation function can be a simple Neural network (RNN or MLP) or an affine transform, $F(h_i) = \mathbf{W}_i h_i + b_i$.

'AGGREGATE' as the aggregation function (sum, mean, max, or min of the neighboring edge message transform), the aggregated message $\mu$ at layer $k$ can be expressed as:

$$\mu_i^{(k)} = \text{AGGREGATE}^{(k)}(\{h_j^{(k)} : \quad j \in \mathcal{N}_i\}) \tag{7}$$

- *Step 3: Update:*
  These aggregated messages update the source edge's features in the GNN Layer. In this updated step, the edge feature vector is combined with the aggregated messages and is executed by simple addition or concatenation:

$$\begin{aligned}
\text{Addition}: \ h_j^{(k+1)} \quad &= \sigma(\Gamma(\Omega(h_j^{(k)}) + \mu_i^{(k)})) \\
\text{Concatenation}: \ h_j^{(k+1)} \quad &= \sigma(\Gamma(\Omega(h_j^{(k)}) \oplus \mu_i^{(k)}))
\end{aligned} \tag{8}$$

where, $\sigma$ is the activation function, $\Omega$ is a simple multi-layer perceptron (MLP), and $\Gamma$ is another neural network that projects the added or concatenated vectors to another dimension. In short, the updating step from the previous layer can be summarized as follows:

$$h_j^{(k+1)} = \text{COMBINE}^{(k)}(h_j^{(k)}, \mu_i^{(k)}) \tag{9}$$

The output of each GNN layer is forwarded as the input to the next GNN layer. After $k$-th GNN layers/iteration, the edge embedding vector at the final layer captures the edge feature information and the graph structure information of all adjacent edges from 1-hop distance to the $k$-th hop distance. The edge feature vector of the 1$^{\text{st}}$ layer is obtained using `NodeVec` [65].

## Proposed GNN framework

Building on the concepts of the GNN presented previously, the proposed GNN framework for estimating edge ranking is presented next. This section introduces the algorithm along with the architecture being proposed. A description of the modified adjacency matrix and its use in edge betweenness ranking is described. This is followed by the details of the edge feature aggregation process in the GNN module. Finally, the details about the ranking loss function are presented.

### Algorithm and the GNN architecture

Fig 5 shows the overall process of calculating the edge ranking using GNN. This framework takes the graph structure—specifically the edge adjacency matrix—and the feature matrix as inputs to estimate the edge ranking vector depicting the importance of each edge in the graph structure. The GNN module is at the core of this procedure, whose inputs are the edge feature matrix and the two variants of the edge adjacency matrix. Starting with initial weights in the GNN module, the edge importance ranking vector of the model is calculated by backpropagating the errors through the GNN layers and then updating the weights iteratively.

**Edge adjacency matrix.** The edge adjacency matrix is not unique for all graph structures. For instance, a pair of non-isomorphic graphs—the three-point star graph $S_3$ and the cycle graph on three vertices $C_3$ have identical edge-adjacency matrices as shown in Fig 6. Hence we introduce two variants for this matrix—modified edge adjacency matrix based on node degree $\tilde{\mathcal{A}}^\varepsilon$ and the modified edge adjacency matrix based on edge weight $\hat{\mathcal{A}}^\varepsilon$. The modified edge

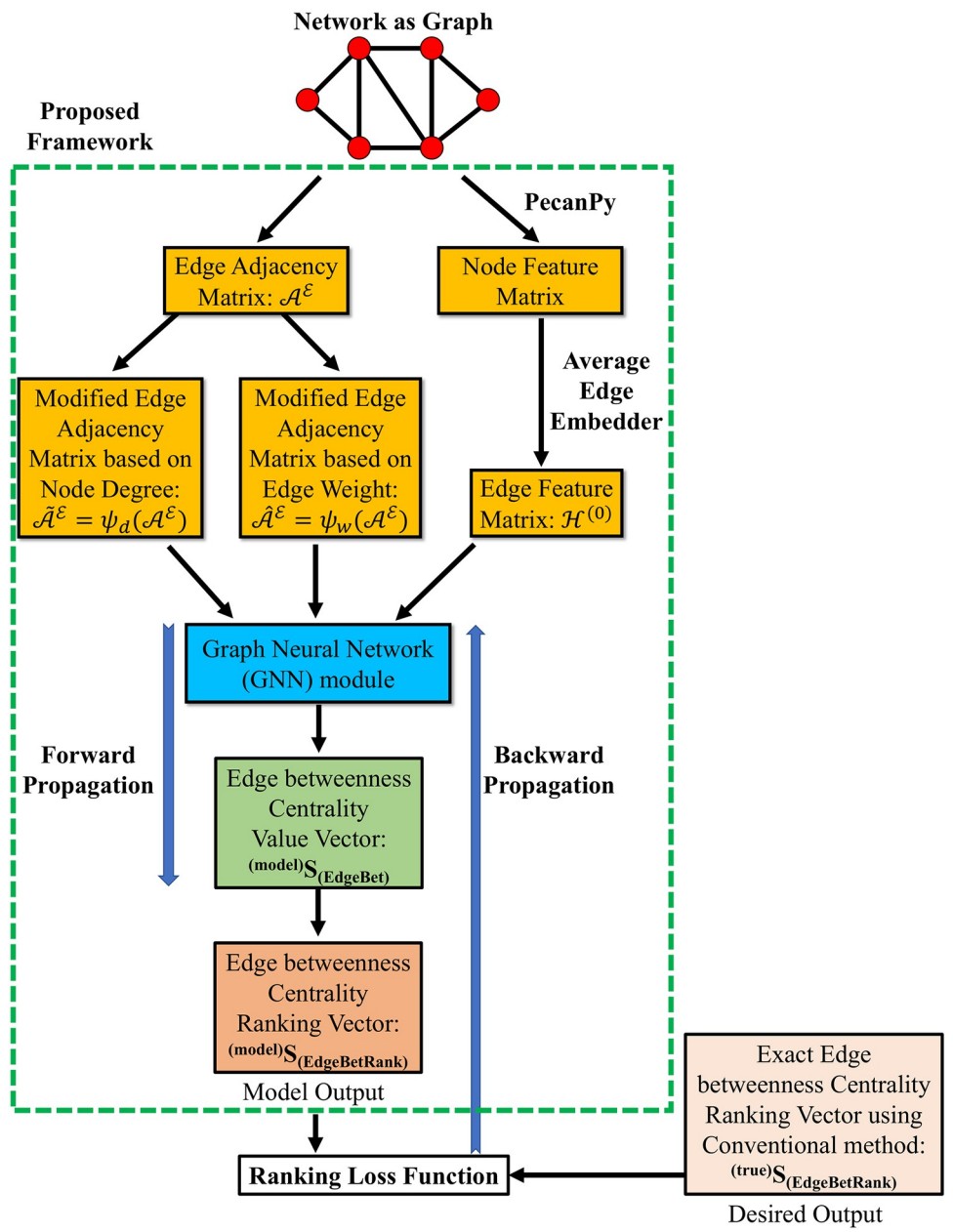

**Fig 5. Proposed framework for edge importance ranking.**

adjacency matrices are obtained from the edge adjacency matrix using the functions $\psi_d$ and $\psi_w$, respectively—shown in detail in Algorithm 1: lines 13-32, and the corresponding example is shown in Fig 7. The edge weights of edges $a$, $b$, $c$, $d$, and $e$ are $(\Omega_{\overrightarrow{a}}, \Omega_{\overleftarrow{a}})$, $(\Omega_{\overrightarrow{b}}, \Omega_{\overleftarrow{b}})$, $(\Omega_{\overrightarrow{c}}, \Omega_{\overleftarrow{c}})$, $(\Omega_{\overrightarrow{d}}, \Omega_{\overleftarrow{d}})$, and $(\Omega_{\overrightarrow{e}}, \Omega_{\overleftarrow{e}})$, respectively, where $(\Omega_{\overrightarrow{i}} \neq \Omega_{\overleftarrow{i}})$ for bidirected graph and $(\Omega_{\overrightarrow{i}} = \Omega_{\overleftarrow{i}} = \Omega_i)$ for an undirected graph, and $i \in (a, b, c, d, e)$. The matrix $\tilde{\mathcal{A}}^{\mathcal{E}}$ is unique to each graph; $\hat{\mathcal{A}}^{\mathcal{E}}$ retains similar features to the original edge adjacency matrix and is non-unique to graph structures.

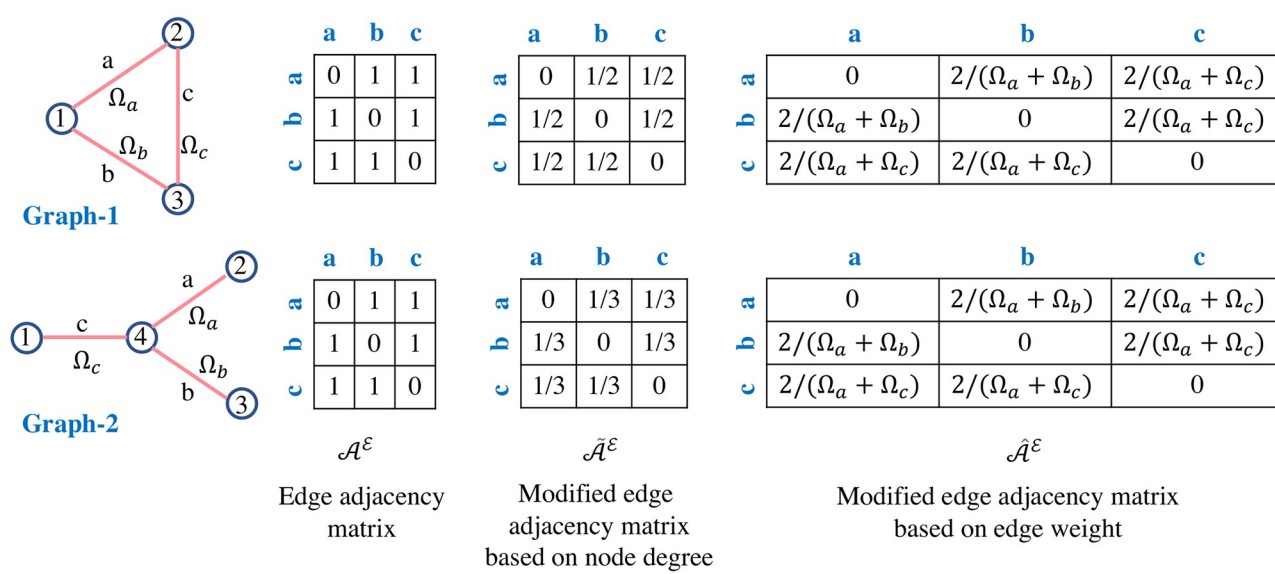

Fig 6. Non-uniqueness of the edge adjacency matrix and the corresponding modifications proposed in the new framework.

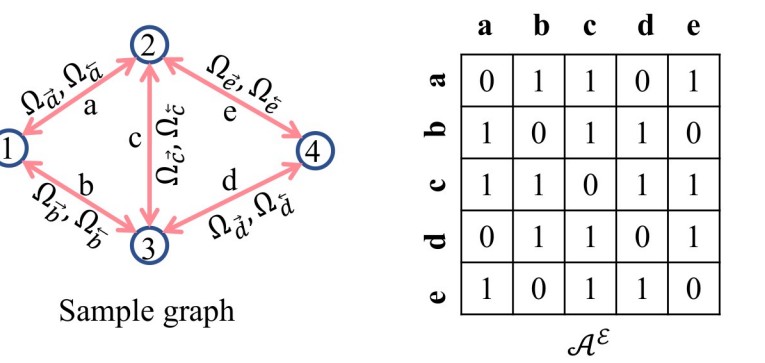

Fig 7. Modified edge adjacency matrices from the edge adjacency matrix of a sample graph network.

**GNN module.** The pseudo-code for the GNN framework and the GNN architecture are shown in Algorithm 1 and Fig 8, respectively. The initial feature for an edge is obtained from the edge embeddings as discussed in the 'Preliminaries' section, denoted as $\mathcal{H}^0$. The features of its $k$-hop neighbors are aggregated at the $k$-th layer. A simple summation of the edge feature vectors is used here for aggregation. At each layer, the feature matrix $\mathcal{H}^0$ is multiplied with the modified adjacency matrices i.e., $\tilde{\mathcal{A}}^{\mathcal{E}}$ and $\hat{\mathcal{A}}^{\mathcal{E}}$. Then, for each edge, the features of the adjacent edges are summed, as shown in Fig 8 and Algorithm 1: line 5-6, with the Leaky-ReLU activation function. The choice of this activation function is not arbitrary and was an outcome of an extensive exercise with different activation functions (details omitted here for brevity).

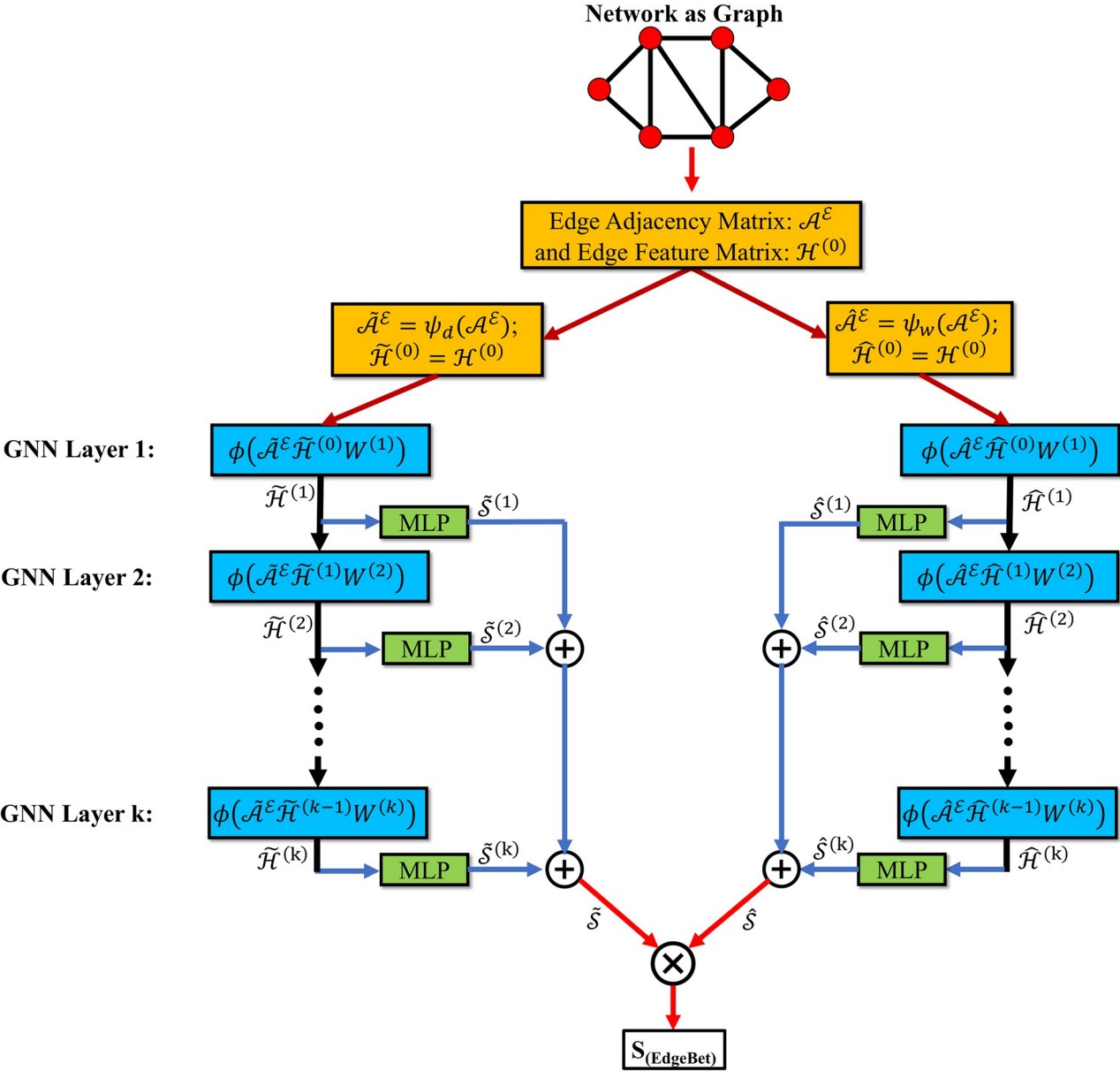

**Fig 8. Proposed Graph Neural Network (GNN) architecture.** This module takes the edge adjacency matrix and the edge feature/embedding matrix obtained from `Node2vec`/`PecanPy` as inputs and calculates the edge importance ranking.

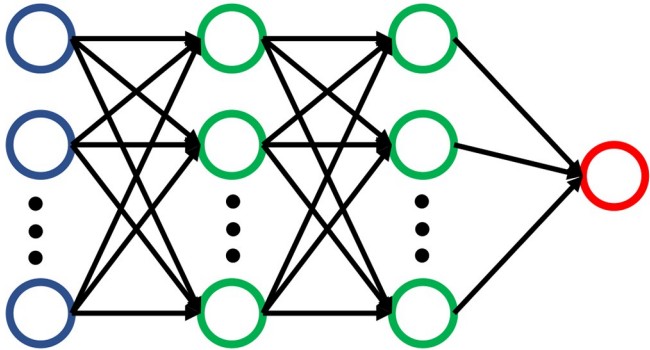

**Fig 9. Multi-Layer Perceptron (MLP) module.**

Subsequently, the aggregated edge features from each GNN layer are mapped to a Multilayer Perceptron (MLP) unit, which outputs a vector of vector space; $\mathbb{R}^{\mathcal{E}}$, and each value of this vector corresponds to each edge of the network as shown in Fig 9 and Algorithm 1: line 7-8. During training, the MLP learns to predict a single score based on the input edge features and the graph connection. Single MLP units are implemented in all layers to output the scores, which are then summed separately as $\tilde{\mathcal{S}}$ and $\hat{\mathcal{S}}$ for the modified adjacency matrices $\tilde{\mathcal{A}}^{\mathcal{E}}$ and $\hat{\mathcal{A}}^{\mathcal{E}}$, respectively. The MLP unit comprises of three fully connected layers and a hyperbolic tangent as the tuned nonlinearity function. The two scores $\tilde{\mathcal{S}}$ and $\hat{\mathcal{S}}$ are multiplied to obtain the final score for each edge as shown in Algorithm 1: line 12. In this architecture, the weights of all the hidden units are initialized using Xavier initialization [72]—which is a standard technique used for weight initialization to ensure the variance of the activations in every layer is identical. Due to the equal variance in every layer, the exploding or vanishing gradient problems are prevented.

**Algorithm 1** GNN based edge betweenness ranking algorithm (forward propagation)

**Input:** Number of Edges $\mathcal{E}$, Edge weight List $\Omega$, unweighted Edge adjacency matrix $\mathcal{A}^{\mathcal{E}}$, Feature matrix $\mathcal{H}^0$, GNN depth $K$, GNN weight matrices $W^{(k)}$

**Output:** Edge betweenness centrality value vector $S_{\text{(EdgeBet)}}$

1: $\tilde{\mathcal{A}}^{\mathcal{E}} \leftarrow \psi_d(\mathcal{A}^{\mathcal{E}})$            ▷ function $\psi_d$ modifies $\mathcal{A}^{\mathcal{E}}$ based on node degree
2: $\hat{\mathcal{A}}^{\mathcal{E}} \leftarrow \psi_w(\mathcal{A}^{\mathcal{E}})$            ▷ function $\psi_w$ modifies $\mathcal{A}^{\mathcal{E}}$ based on edge weight
3: $\tilde{\mathcal{H}}^{(0)} = \hat{\mathcal{H}}^{(0)} = \mathcal{H}^0$
4: **for** $k = 1, \cdots, K$ **do**
5:     $\tilde{\mathcal{H}}^{(k)} \leftarrow \phi(\tilde{\mathcal{A}}^{\mathcal{E}} \tilde{\mathcal{H}}^{(k-1)} W^{(k)})$
6:     $\hat{\mathcal{H}}^{(k)} \leftarrow \phi(\hat{\mathcal{A}}^{\mathcal{E}} \hat{\mathcal{H}}^{(k-1)} W^{(k)})$            ▷ $\phi$ is the activation function
7:     $\tilde{\mathcal{S}}^{(k)} \leftarrow \text{MLP}(\tilde{\mathcal{H}}^{(k)})$
8:     $\hat{\mathcal{S}}^{(k)} \leftarrow \text{MLP}(\hat{\mathcal{H}}^{(k)})$            ▷ MLP is the multi-layer perceptron
9: **end for**
10: $\tilde{\mathcal{S}} \leftarrow \sum_{k=1,...,K} |\tilde{\mathcal{S}}^{(k)}|$
11: $\hat{\mathcal{S}} \leftarrow \sum_{k=1,...,K} |\hat{\mathcal{S}}^{(k)}|$
12: $S_{\text{(EdgeBet)}} \leftarrow \tilde{\mathcal{S}} \times \hat{\mathcal{S}}$

```
13: function ψ_d (A^ε)
14:    Ã^ε = zeros(ε, ε)
15:    for  i = 1, ⋯, ε do
16:      for  j = i + 1, ⋯, ε do
17:         Ã^ε(i, j) = ────────────────────────────────────────────
                        Degree of the node connecting edge i and j
18:         Ã^ε(j, i) = Ã^ε(i, j)
19:      end for
20:    end for
21:    return  Ã^ε
22: end function
23: function ψ_w (A^ε)
24:    Â^ε = zeros(ε, ε)
25:    for  i = 1, ⋯, ε do
26:      for  j = 1, ⋯, ε do
27:         Â^ε(i, j) = ────────────────
                        (Ω_{→i} + Ω_{→j})/2
28:         Â^ε(j, i) = ────────────────
                        (Ω_{←i} + Ω_{←j})/2
29:      end for
30:    end for
31:    return  Â^ε
32: end function
```

## Loss function

A ranking loss function is used to estimate the loss due to the differences in the ranking predicted by the proposed model compared to the target EBC ranking. Such ranking loss functions have previously been used for recommendation systems to rank or rate products or users [73]. The margin ranking loss function is defined as follows:

$$
\begin{aligned}
L(S_i^{(\text{model})}, S_i^{(\text{true})}, y) \quad &= \text{Max}\left(0, -y \cdot \left(S_i^{(\text{model})} - S_i^{(\text{true})}\right) + \text{Margin}\right) \\
y \quad &= \begin{cases} 1, & \text{if } S_i^{(\text{model})} \text{ should be ranked higher than } S_i^{(\text{true})} \\ -1, & \text{if } S_i^{(\text{true})} \text{ should be ranked higher than } S_i^{(\text{model})} \end{cases}
\end{aligned} \tag{10}
$$

where, $S_i^{(model)}$ is the predicted ranking-score, and $S_i^{(true)}$ is the EBC score obtained using Brandes Algorithm [63] as shown in Eq 3. In this study, the margin value is set to 1 to allow some flexibility.

## Conventional edge ranking algorithm for comparison purposes

In this section, we describe two vulnerability-based network indices [74, 75] which have been used for ranking important roads: (a) network efficiency-based measure and (b) probabilistic measure of distance between networks. These two metrics are summarized as follows:

**Network efficiency-based vulnerability measure.** Considering a connected network **G**, the network efficiency metric [74] is given by:

$$
E(\mathbf{G}) = \frac{1}{N(N-1)} \sum_{i \neq j \in \mathbf{G}} \frac{1}{d_{ij}}; \qquad d_{ij} = \begin{cases} \ell_{ij} & \text{if path}(i, j) \text{exists} \\ 0 & \text{if } i = j \\ \infty & \text{if path}(i, j) \text{does not exists} \end{cases} \tag{11}
$$

where, $\ell_{ij}$ is the shortest path between node $i$ and $j$, and $N$ is the total number of nodes in the network. The metric $E(\mathbf{G})$ represents an average across all node pairs of the reciprocals of the node pair distance. A large value of $E(\mathbf{G})$ corresponds to faster travel times in the network. If any link in the network is disconnected, which can occur following a disruption, $E(\mathbf{G})$ reduces. Hence, edge importance ranking can be calculated based on the network efficiency reduction for edge interruption –removing a high-ranked edge will lead to large network efficiency reduction. Therefore, the network vulnerability [76] can be defined as the relative drop in network efficiency caused by the removal of an edge $i$:

$$C_\Delta^E(i) = \frac{E(\mathbf{G}) - E(\mathbf{G} \setminus \{i\})}{E(\mathbf{G})} \tag{12}$$

here, $\mathbf{G} \{i\}$ denotes the network $\mathbf{G}$ without the edge $i$.

**Probabilistic measure of distance between networks.** This vulnerability measure is based on node-to-node distance distribution whose density values are the number of nodes that are connected at a distance $k$, where $a \leq k \leq b$, with each node $i = 1, \cdots, N$ of the graph $\mathbf{G}$. This continuous distance distribution for each node is given by:

$$P_i(a \leq k \leq b) = n_{i,k}. \tag{13}$$

The un-normalized distance distribution over the whole graph is given by,

$$\tilde{P}_G(k) = \sum_{i=1}^{n} P_i(a \leq k \leq b) = \sum_{i=1}^{n} n_{i,k}. \tag{14}$$

The normalized distance distribution over the whole graph is then given by

$$P_G(k) = \frac{\tilde{P}_G(k)}{\int_{\mathcal{I}} \tilde{P}_G(k) \, \mathrm{d}k}. \tag{15}$$

$\mathcal{I}$ is the interval where the continuous probability density function (PDF) is defined. Such PDFs can be obtained for the original uninterrupted graph $\mathbf{G}$ and the interrupted graph $\mathbf{G} \{i\}$, denoted as $P_G(k)$ and $P_{G\{i\}}(k)$, respectively. After deleting the $i$th edge, the shortest path between nodes should theoretically increase for some or all cases, which means that the histogram for the interrupted graph $P_{G\{i\}}(k)$ will shift to the right compared to $P_G(k)$ with some change in the distribution shape. This distance between the distributions will determine the vulnerability of the network to edge interruption.

For comparing two probability distributions, several distance and divergence measures, like Kullback-Leibler (KL) divergence and Jensen-Shannon (JS) divergence, can be used. A robust distance measure for network vulnerability analysis is the Wasserstein distance [75] or the Earth Mover (EM) distance, given by:

$$W(P_G(k), P_{G\setminus\{i\}}(k)) = \inf_{\gamma \in \Gamma(P_G(k), P_{G\setminus\{i\}}(k))} \mathbb{E}_{(x,y)\sim\gamma}[||x - y||]. \tag{16}$$

Here, $\Gamma(P_G(k), P_{G\{i\}}(k))$ represents the set of all joint distribution $\gamma(x, y)$ whose marginals are $P_G(k)$ and $P_{G\setminus\{i\}}(k))$ respectively. Intuitively, $\gamma(x, y)$ indicates the amount of mass transported from $x$ to $y$ to transform the distribution $P_G(k)$ into the distribution $P_{G\setminus\{i\}}(k)$.

Both of these aforementioned vulnerability measures need to calculate the `All pair shortest path` for all the cases when one of the edges is interrupted. The fastest way to calculate the `All pair shortest path` is using Johnson's Algorithm [77], and it has a computational complexity of $\mathcal{O}(|\mathcal{V}||\mathcal{E}| + |\mathcal{V}|^2 \log |\mathcal{V}|)$. Hence, the vulnerability-based edge

ranking procedure will have a computational complexity of $\mathcal{O}(|\mathcal{V}||\mathcal{E}|^2 + |\mathcal{E}||\mathcal{V}|^2 \log |\mathcal{V}|)$. We found that these vulnerability-based edge ranking techniques also correlated well with the Edge Betweenness Centrality-based edge ranking technique (Brandes's method).

## Results

Experimental results for both the synthetic and real-world cases with undirected and bidirected cases are presented in this section. First, details of training for the proposed architecture are discussed. Then, the network performance on synthetic graphs of two real-world transportation networks is presented.

### Training

**Hardware and software configuration.** In terms of computing resources, the experiments were conducted on a dedicated computer and the details are shown in Table 1.

Following best practices, the graph datasets are divided into training ($\approx$80%), validation ($\approx$10%), and testing ($\approx$10%) datasets [78]. The EBC ranking is calculated using the Brandes's method [40] for all graphs in the training and testing datasets. These rankings are used as target vectors for training the GNN. The test graphs are not used for training; the model learns to map edge features and importance via MLP to the ranking scores. Training and testing graphs contain variable numbers of nodes and edges, and the GNN is trained and tested on the same type of synthetic graph. The performance of the GNN model is compared with the EBC and other edge ranking algorithms, and the evaluation accuracy is measured using Spearman's rank correlation coefficient.

**Evaluation metrics—Spearman's rank correlation coefficient.** Spearman's rank correlation coefficient [79] is defined as the ratio of the covariance of two rank variables and the product of their standard deviations. With $n$ number of observations, the $n$ raw scores of variables $x_i$ and $y_i$ are transformed to the ranks $R(x_i)$ and $R(y_i)$ for the joint random variables $X$ and $Y$. Then the Spearman's rank correlation coefficient $\rho$ is expressed as follows:

$$\rho_s = \frac{\text{cov}(R(X), R(Y))}{\sigma_{R(X)}\sigma_{R(Y)}} \tag{17}$$

where, $\text{cov}(R(X), R(Y))$ is the covariance of the rank variables; $\sigma_{R(X)}$ and $\sigma_{R(Y)}$ are the standard deviations of the rank variables. If all the $n$ ranks are distinct integers, this can be simplified as:

$$\rho_s = 1 - \frac{6 \sum r_i^2}{n(n^2 - 1)} \tag{18}$$

where $r_i = R(x_i) - R(y_i)$ is the difference between the two ranks of each observation, and $n$, is the number of observations. The range of Spearman's coefficient $\rho_s$ is $-1 \le \rho_s \le 1$; $\rho_s = 1, -1$, and 0 denote perfectly positive, perfectly negative, and no correlation, respectively.

**Table 1. Hardware and software configuration used for experiments.**

| CPU model and speed | Intel Core i9-10940X CPU @ 3.30 GHz |
|---|---|
| Available Hard disk | 1 TB |
| Available RAM | 64 GB |
| GPU type and specification | NVIDIA GeForce RTX 3090—24 GB |
| Programming | Python 3.7, Matlab 2022a |
| Deep Learning framework | PyTorch, Numpy, Scipy, CUDA 11.6 |
| GNN framework | NetworkX, Node2Vec, Pecanpy, Gensim |

**Hyper-parameters.** A model size of 10000 (number of edges in the largest graph) is used here as a typical example for a medium-sized urban transportation network in the U.S. The size of the edge adjacency matrix (input) size is fixed at 10000 edges (Fig 8), and smaller graphs can be accommodated by populating only the upper left portion of this matrix, with zeros elsewhere. The network is trained using the ADAM (Adaptive Moment Estimation) optimizer, which is a variant of the stochastic gradient descent (SGD) algorithm [80]. The hyper-parameter training was performed with a learning rate of 0.0005 and a dropout ratio of 0.3. The number of epochs used for training is 50, and the number of hidden neurons (embedding dimension) and the number of GNN layers () were optimized. The experiments use an embedding dimension of 256 and 5 layers. The edge features are obtained using `PecanPy` with the feature vector of length 256. BFS approach with $p = 1$ and $q = 2$ is used to search the shortest path. Since the calculation of the ranking loss function requires a comparison of edge pair ranking for all possible combinations of edge pairs, i.e. $\binom{|\mathcal{E}|}{2}$ for $|\mathcal{E}|$ edges, instead, 20 times the number of edges are randomly sampled [57].

## Performance on synthetic networks

Three synthetic random graphs are used to evaluate the performance of the proposed method: (a) Erdős—Rényi variant-I [81] i.e., $G_{np}$—graphs containing $n$ nodes (fixed number) where each edge $(u, v)$ appears independent and identically distributed with probability $p$; (b) Erdős—Rényi variant-II [81] i.e., $G_{nm}$—graphs containing $n$ nodes (fixed number) and $m$ edges, with edges uniformly connected to random nodes. Unlike Erdős—Rényi variant-I, the number of edges in Erdős—Rényi variant-II are fixed; and (c) Watts–Strogatz model [82]—a random graph generation model that produces graphs with small-world properties such as local clustering and average shortest path lengths. The small world random graph has been used in applications such as electric power grids, networks of brain neurons and airport networks [83]. Both undirected and bidirected cases are considered in each synthetic graph type.

**Synthetic graph generation parameters.** Table 2 shows the synthetic graph generation parameters. Here, $\mathcal{U}\{a, b\}$ and $\mathcal{U}[a, b]$ represent discrete and continuous uniform distributions between the ranges $a$ and $b$, respectively. Here, $\llcorner \cdot \urcorner$ denotes the nearest integer function. $\mathcal{T}(c, d, e)$ denotes a triangular continuous probability distribution with a lower limit left $c$, peak at mode $d$, and upper limit right $e$. With the graph generation parameters chosen

**Table 2. Generation parameters of the synthetic graphs.**

| Synthetic Graph Type | Generation Parameters | | |
|---|---|---|---|
| Erdős—Rényi-I (ER-I) or $G_{np}$ random | # nodes | | $\mathcal{U}\{1000, 5000\}$ |
| | Probability of edge creation | | 1.2/(Nos of nodes -1) |
| | Edge weights (undirected) | | $\mathcal{U}[0, 100]$ |
| | Edge weights (bidirected) | | $\Omega_{\vec{i}} = \mathcal{U}[0, 100]; \Omega_{\vec{i}}^{-} = \mathcal{T}(1, 1, 5) \times \Omega_{\vec{i}}$ |
| Erdős—Rényi-II (ER-II) or $G_{nm}$ random | # nodes | | $\mathcal{U}\{1000, 5000\}$ |
| | # edges | | $\mathcal{U}[1.4, 1.6] \times$ Nos of nodes |
| | Edge weights | | $\mathcal{U}[0, 100]$ |
| | Edge weights (bidirected) | | $\Omega_{\vec{i}} = \mathcal{U}[0, 100]; \Omega_{\vec{i}}^{-} = \mathcal{T}(1, 1, 5) \times \Omega_{\vec{i}}$ |
| Watts-Strogatz (WS) or Small world | # nodes | | $\mathcal{U}\{2000, 4000\}$ |
| | Mean degree | | 4 |
| | Probability of edge rewiring | | 0.5 |
| | Edge weights | | $\mathcal{U}[0, 100]$ |
| | Edge weights (bidirected) | | $\Omega_{\vec{i}} = \mathcal{U}[0, 100]; \Omega_{\vec{i}}^{-} = \mathcal{T}(1, 1, 5) \times \Omega_{\vec{i}}$ |

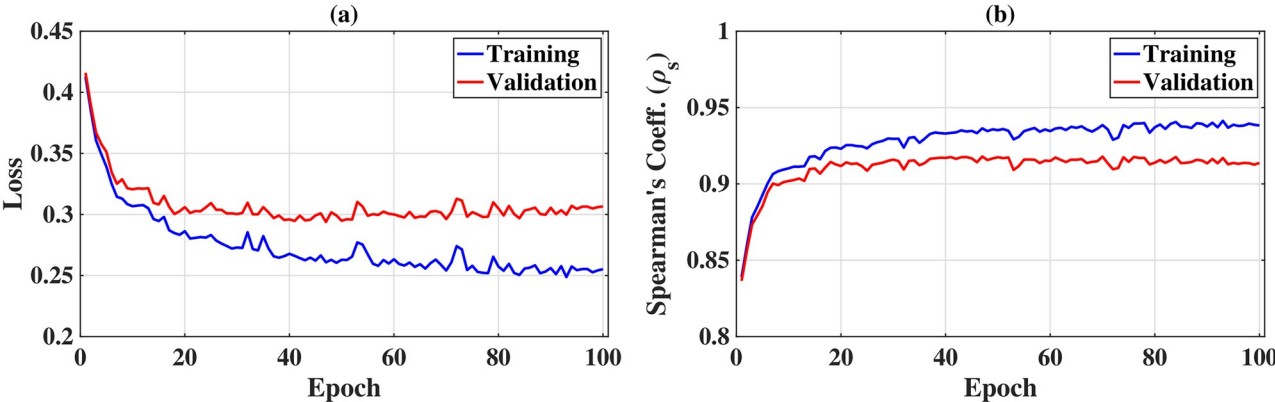

**Fig 10. Evolution in the learning phase.** (a) Training and validation loss over epochs and (b) Evaluation metric (for training and validation data) over epochs.

arbitrarily for this case study, we generate 1000 training graphs, 100 validation graphs, and 100 test graphs for each case.

**Model training time.** Fig 10(a) displays the evolution of the evaluated training and validation marginal ranking loss for each epoch. The progression of Spearman's rank correlation coefficient over epochs for both training and validation data is also depicted in Fig 10(b). It is observed from Fig 10(a) that after 50 epochs, the training loss continues to decrease while the validation loss starts to increase—indicating the 'overfitting' phenomenon in deep learning [84]. As 'early stopping' is considered one of the robust ways to prevent overfitting by halting training when the validation loss ceases to improve, the training process was terminated at 50 epochs. Each epoch took approximately 175 seconds to train on average, resulting in a total training time of $\approx$2.43 hours.

**Performance and comparison with other methods.** The Spearman correlation for the three types of synthetic graphs including undirected and bidirected cases is shown in Table 3. The first two rows, 'training' and 'testing', show the comparison or correlation between the edge importance ranking obtained from EBC and GNN. The results also compare the edge importance ranking obtained from the network-efficiency-based vulnerability measure (Eff. Vul.) and the vulnerability based on the probabilistic distance measure between networks (Prob. Vul.) with the proposed GNN framework in the subsequent rows. The high correlation observed in the results shows that the proposed GNN framework is able to predict the edge importance rank for various examples studied. The detailed ranking score statistics in the form of the Box and Whisker plot are also shown in Fig 11. The standard deviations for the estimated scores are small, indicating that the method is also robust.

**Table 3. Spearman's coefficient on synthetic graphs.**

| Graph | ER-I | | ER-II | | WS | |
|---|---|---|---|---|---|---|
| | **Undirected** | **Bidirected** | **Undirected** | **Bidirected** | **Undirected** | **Bidirected** |
| Training | 0.94 ± 0.005 | 0.90 ± 0.006 | 0.93 ± 0.006 | 0.90 ± 0.008 | 0.94 ± 0.003 | 0.90 ± 0.005 |
| Testing | 0.92 ± 0.009 | 0.88 ± 0.009 | 0.92 ± 0.012 | 0.88 ± 0.014 | 0.92 ± 0.005 | 0.87 ± 0.006 |
| Eff. Vul. | 0.90 ± 0.010 | 0.84 ± 0.021 | 0.88 ± 0.011 | 0.85 ± 0.015 | 0.82 ± 0.008 | 0.85 ± 0.011 |
| Prob. Vul. | 0.85 ± 0.018 | 0.77 ± 0.031 | 0.82 ± 0.020 | 0.79 ± 0.024 | 0.78 ± 0.012 | 0.81 ± 0.014 |

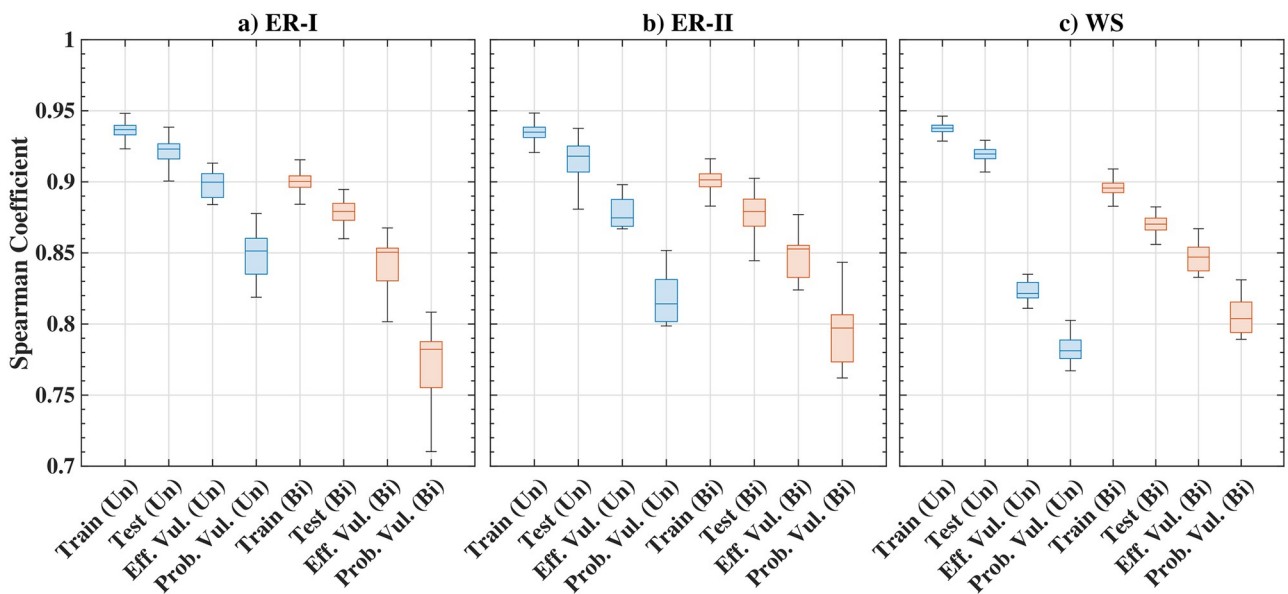

**Fig 11. Results of edge ranking score distributions for synthetic graphs; box and whisker plot shows the median, the lower and upper quartiles, any outliers (calculated using the interquartile range), and the minimum and maximum values that are not outliers.**

**Inference speed.** The proposed framework's inference speed combines the latencies of GNN and `PecanPy`. While the computational overhead with the inference part of the GNN is on the order of milliseconds, `PecanPy` has a relatively large computational overhead. Hence, the complexity of the overall framework is governed by the complexity of the `PecanPy`, which has a time complexity of $\mathcal{O}(|\mathcal{V}|)$ [85]. Fig 12 compares the proposed GNN-based edge ranking approach with the EBC-based and vulnerability-based edge ranking techniques in a semi-log plot. Vulnerability based ranking approach is extremely time-consuming for large graphs (computational complexity of $\mathcal{O}(|\mathcal{V}||\mathcal{E}|^2 + |\mathcal{E}||\mathcal{V}|^2 \log |\mathcal{V}|)$); hence we show the results only for graphs with a maximum of 1,000 nodes. Also, beyond the graph size of about 2,000 nodes, the proposed GNN method outperforms Brandes' EBC-based ranking method (computational complexity of $\mathcal{O}(|\mathcal{V}||\mathcal{E}| + |\mathcal{V}|^2 \log |\mathcal{V}|)$). It should be noted that these comparisons apply to the inference phase of the GNN and do not include the training phase. These results underscore the advantage of the proposed GNN method for large graphs compared to the conventional method. Results show that the proposed method can generate results significantly faster in the inference phase while accompanied only by a slight reduction in accuracy, which can be very beneficial in emergency response scenarios.

Two variants of edge-adjacency matrices are used in the proposed framework, as shown in Fig 8. Using the edge adjacency matrix, or the modified edge adjacency matrices, individually results in Spearman's values between 0.48-0.91, while combining $\tilde{\mathcal{A}}^{\mathcal{E}}$ and $\hat{\mathcal{A}}^{\mathcal{E}}$ outperforms these cases, resulting in a value of 0.92.

## Ablation studies on road networks

Following, the performance of the proposed GNN framework is evaluated on six transportation networks around the world through ablation studies. 1) the U.S. inter-state highway network, 2) Minnesota state road Network (US), 3) Aachen city transportation network (Germany), 4) Edinburgh city transportation network (Scotland), 5) Road network of country

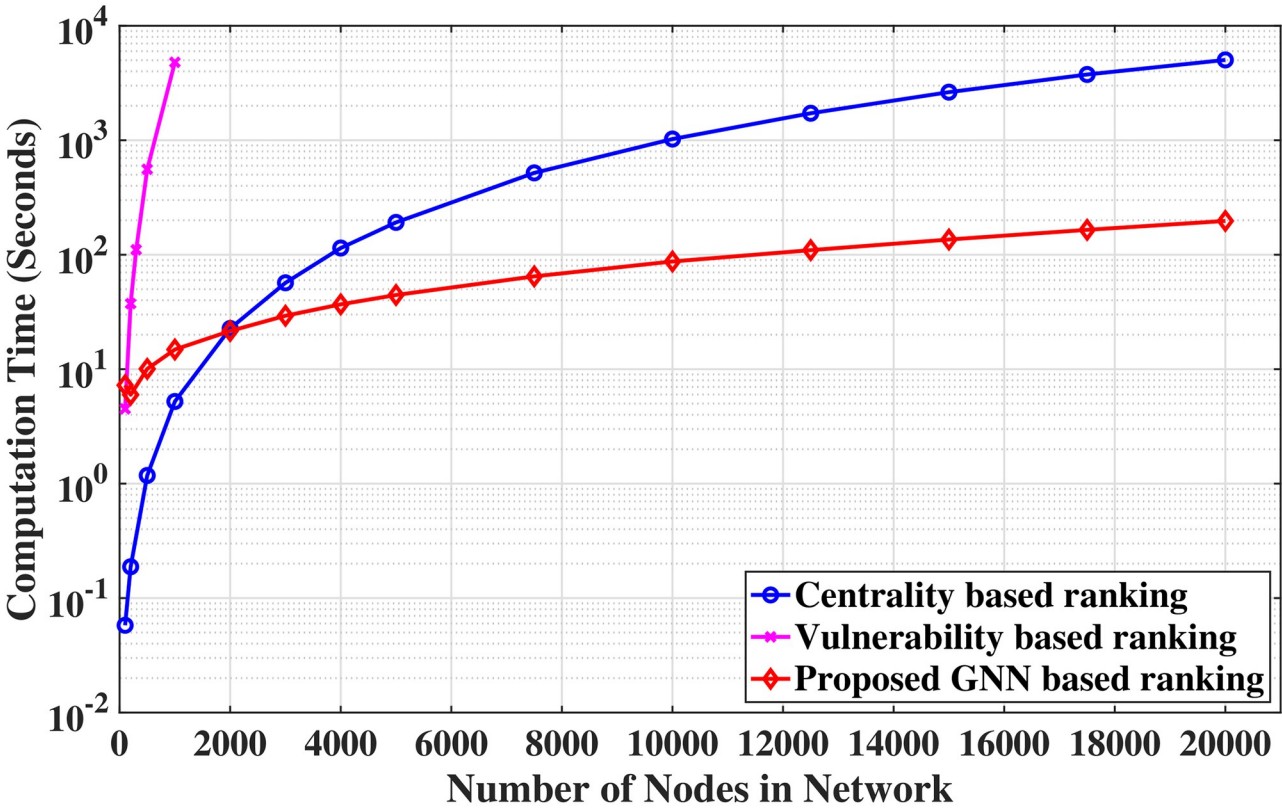

**Fig 12. Comparison of times required for computation for edge ranking of graphs between the conventional methods and the proposed GNN-based approach.**

of Luxembourg (Europe), and 6) Santa Barbara city transportation network (US). In our analysis, we take into account both undirected and bidirected scenarios. In the undirected case, we assume equal travel times in both directions. This following ablation study simulates the edge importance ranking due to a dynamic change in the network parameters resulting from construction activities or functional factors such as travel time. Two scenarios are considered in this study:

(i) Case I: In this case, the effects of minor congestion or interruption of traffic on roads are simulated through random perturbations in the edge weights according to $r \times \Omega_i$ where $\Omega_i$ is the weight of edge $i$, and the values of $r$ are sampled from a uniform distribution $\mathcal{U}[0.8, 1.2]$. The number of nodes and edges for the graphs remains unchanged.

(ii) Case II: In this case, major interruption scenarios rendering a small number of edges inoperable, such as from accidents or natural disasters, are simulated. In addition to the edge weights specified previously, i.e., $r \times \Omega_i$ with $r \in \mathcal{U}[0.8, 1.2]$, the number of edges in the graph is modified as well. The number of edges is sampled from a discrete uniform distribution $\mathcal{U}[0.99, 1] \times \mathcal{E}$—which is based on a maximum of 1% edge deletion from the original network. Here, $\cdot$ denotes the nearest integer function. While the 1% edge deletion number is arbitrarily assumed for the ablation study, this is not far from reports from previous events such as from a 20-year return period event resulting in 0.6%-0.7% loss in the road inventory [86].

**Table 4. Generation parameters of the transportation networks; for each network, the training, testing, and validation data are generated from a range of edge weights as follows: $\Omega_i = 0.5 \times (\mathbb{G}_i^{\rightarrow} + \mathbb{G}_i^{\leftarrow}) \times \mathcal{U}[0.8, 1.2]$ for undirected graphs, and $\Omega_i^{\rightarrow} = \mathbb{G}_i^{\rightarrow} \times \mathcal{U}[0.8, 1.2]$ and $\Omega_i^{\leftarrow} = \mathbb{G}_i^{\leftarrow} \times \mathcal{U}[0.8, 1.2]$ for bidirected graphs.**

| Transportation Network | Generation Parameters | Case I | Case II |
|---|---|---|---|
| US Highway, USA | # nodes | 997 | 997 |
| | # edge | 1490 | $\mathcal{U}[0.99, 1] \times 1490$ |
| Minnesota, USA | # nodes | 2640 | 2640 |
| | # edge | 3302 | $\mathcal{U}[0.99, 1] \times 3302$ |
| Aachen, Germany | # nodes | 791 | 791 |
| | # edge | 1498 | $\mathcal{U}[0.99, 1] \times 1498$ |
| Edinburgh, Scotland | # nodes | 864 | 864 |
| | # edge | 1497 | $\mathcal{U}[0.99, 1] \times 1497$ |
| Luxembourg, Europe | # nodes | 692 | 692 |
| | # edge | 1495 | $\mathcal{U}[0.99, 1] \times 1495$ |
| Santa Barbara USA | # nodes | 821 | 821 |
| | # edge | 1499 | $\mathcal{U}[0.99, 1] \times 1499$ |

Data for the U.S. system is obtained from the Environmental Systems Research Institute, Inc. (ESRI) [87]. It includes approximately 60k miles of interstate highways in the U.S. After condensing tiny segments using geographic clustering [88], the resulting graph contains 997 nodes (road junctions) and 1490 edges (streets). The network information of a much larger network, Minnesota state road network, is obtained from the network repository [89], which contains 2640 nodes (road junctions) and 3302 edges (roads). The remaining four transportation network datasets are acquired from Pyrosm, a Python library for OpenStreetMap data [90]. These datasets also underwent the same condensation process, yielding graphs with approximately 1500 edges each. The edge weights are assigned from travel times obtained through the Google Distance Matrix API [91], and denoted as $\mathbb{G}_i^{\rightarrow}$ and $\mathbb{G}_i^{\leftarrow}$ for bidirectional travel time on edge $i$. Table 4 presents the generation parameters for the GNN operation for all the real-life transportation networks.

The US transportation network is shown in Fig 13(a). The EBC scores obtained for this network are shown in Fig 13(b)–roads highlighted in red are the most critical roads, i.e., changes to these segments such as edge weights or addition and deletion, will impact the overall network significantly. In the context of GNN training, validation, and testing, the primary

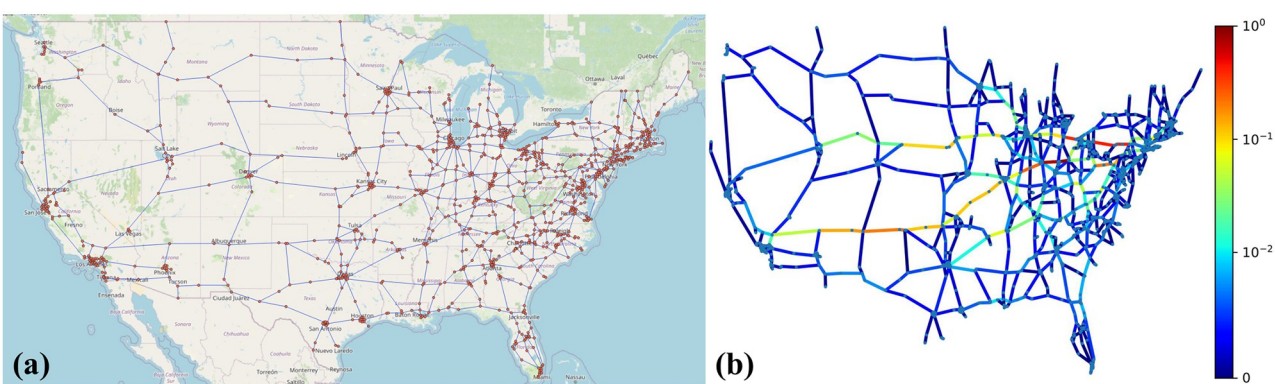

**Fig 13. EBC score calculated for the U.S. freeway network.** (a) Road network shown on the map, (b) Normalized EBC score.

**Table 5. Spearman's coefficient on us freeway network.**

| Graph | Case I | | Case II | |
|---|---|---|---|---|
| | Undirected | Bidirected | Undirected | Bidirected |
| Train | 0.92 ± 0.009 | 0.92 ± 0.008 | 0.89 ± 0.015 | 0.91 ± 0.014 |
| Test | 0.89 ± 0.014 | 0.92 ± 0.009 | 0.89 ± 0.017 | 0.90 ± 0.015 |
| Eff. Vul. | 0.77 ± 0.004 | 0.75 ± 0.006 | 0.76 ± 0.010 | 0.74 ± 0.008 |
| Prob. Vul. | 0.83 ± 0.006 | 0.81 ± 0.006 | 0.83 ± 0.011 | 0.80 ± 0.010 |

distinction among simulations lies in the change of edge weights, reflecting changes in travel time due to factors such as congestion (Case-I) and a deletion of edges (e.g., due to road closures) resulting from catastrophic events (Case-II). More details regarding this change can be found in Table 4. Spearman correlation is calculated for both previously described cases, considering both undirected and bidirected graphs, to assess perturbation scenarios. The results of these evaluations are detailed in Fig 5. The ranking score statistics in Fig 14 and Table 5 show a high Spearman score, underscoring the effectiveness of our GNN approach.

For better visualization purposes, we compare the ranking of edges/roads using our proposed GNN with EBC, as depicted in Fig 15. This comparison plot demonstrates that the trained GNN architecture, using EBC-based ranking as a surrogate target, effectively mimics the performance of EBC-based ranking with much less computational cost, leveraging its capacity to generalize from a broad range of network parameter changes and disruption events. Hence, GNN-based ranking is useful for a fast approximation of critical road segments (Fig 12), making it beneficial for the application of hazard simulation and rapid decision-

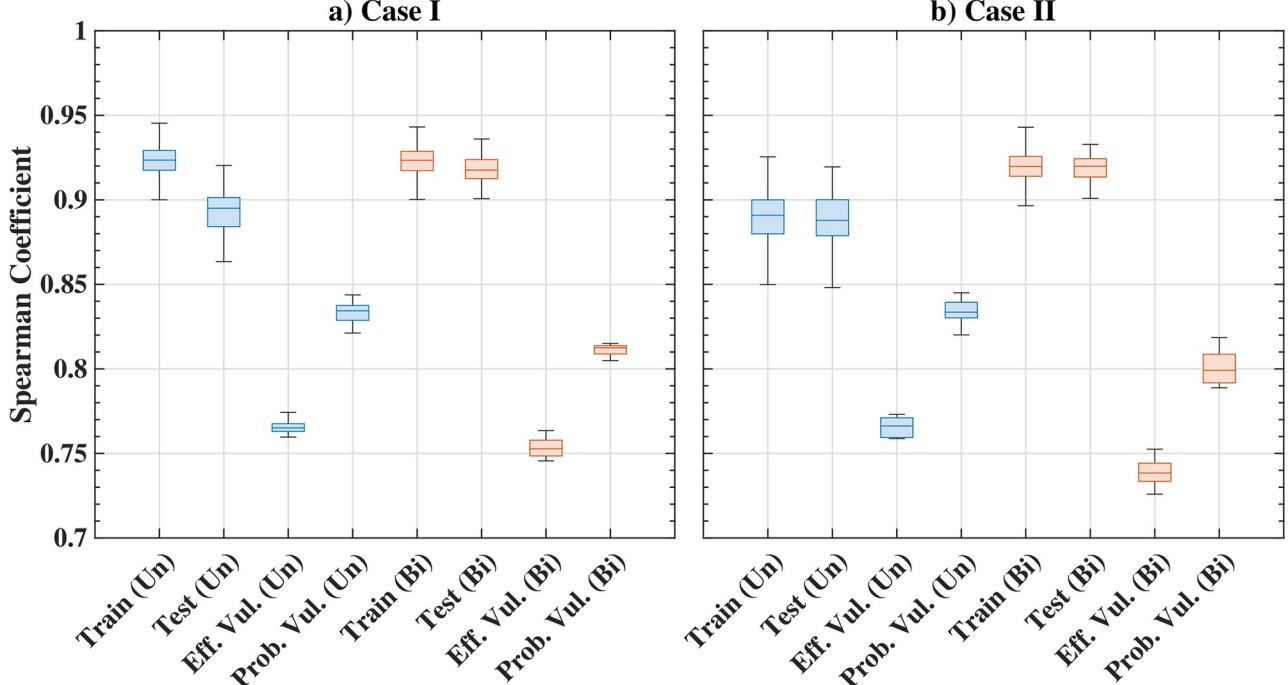

**Fig 14. Edge ranking score distributions for US freeway Network.** The Box and Whisker plot shows the median, the lower and upper quartiles, any outliers (calculated using the interquartile range), and the minimum and maximum values that are not outliers.

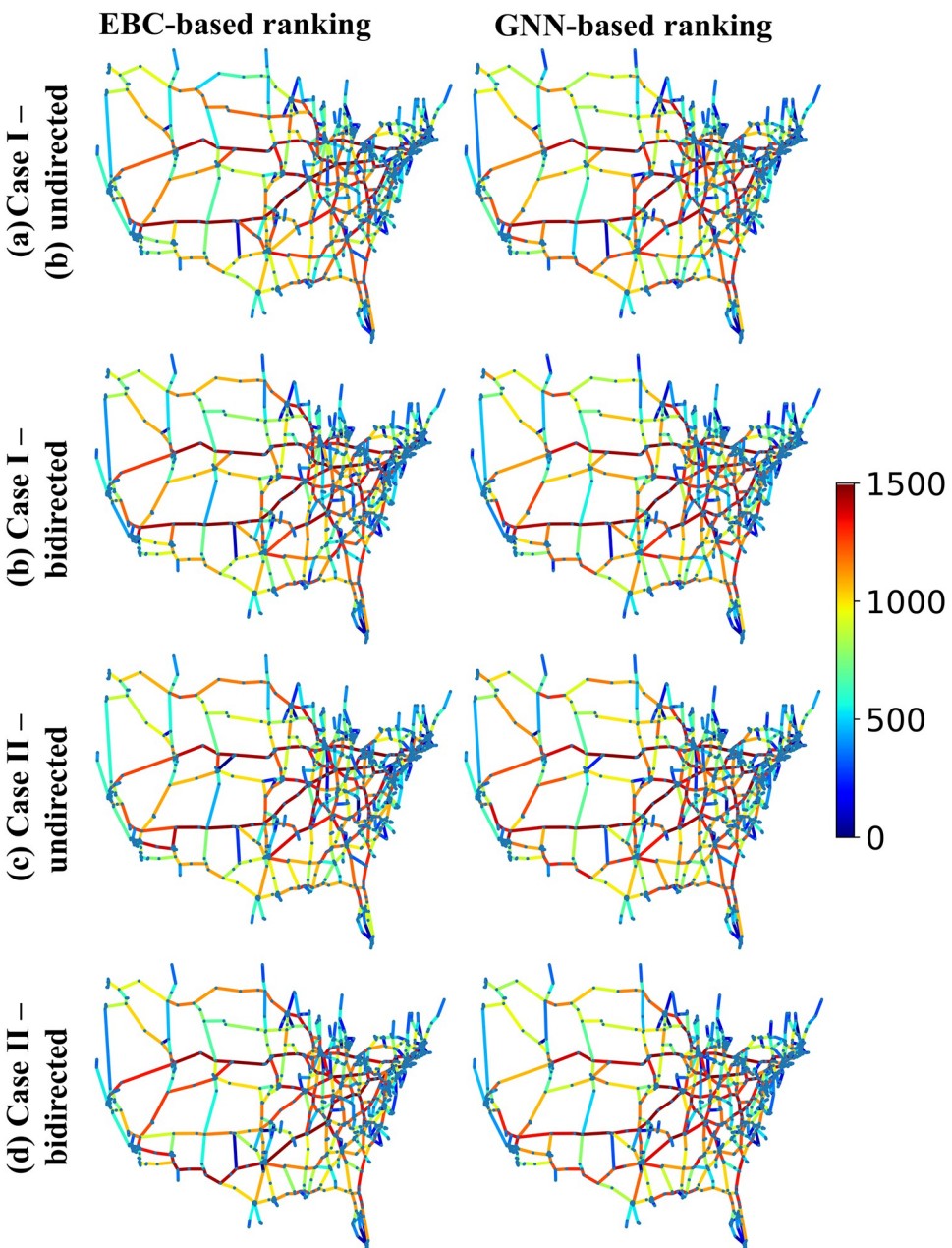

**Fig 15. Comparison between EBC based road ranking and the proposed GNN-based road ranking of US highway network for different scenarios; the color bar shows the road ranking.**

making in emergency situations for large networks. It is also noteworthy that while the GNN was trained on EBC-based ranking, it has not been explicitly trained on network-efficiency-based vulnerability measure-based ranking (Eff. Vul.) and the vulnerability based on the probabilistic distance measure between networks based ranking (Prob. Vul.). Despite this, Fig 14 and Table 5 demonstrate that the GNN is capable of approximating the ranking based on these more computationally intensive quantities. This finding underscores the versatility of our proposed method and its potential applicability in scenarios where directly calculating vulnerability metrics might be impractical.

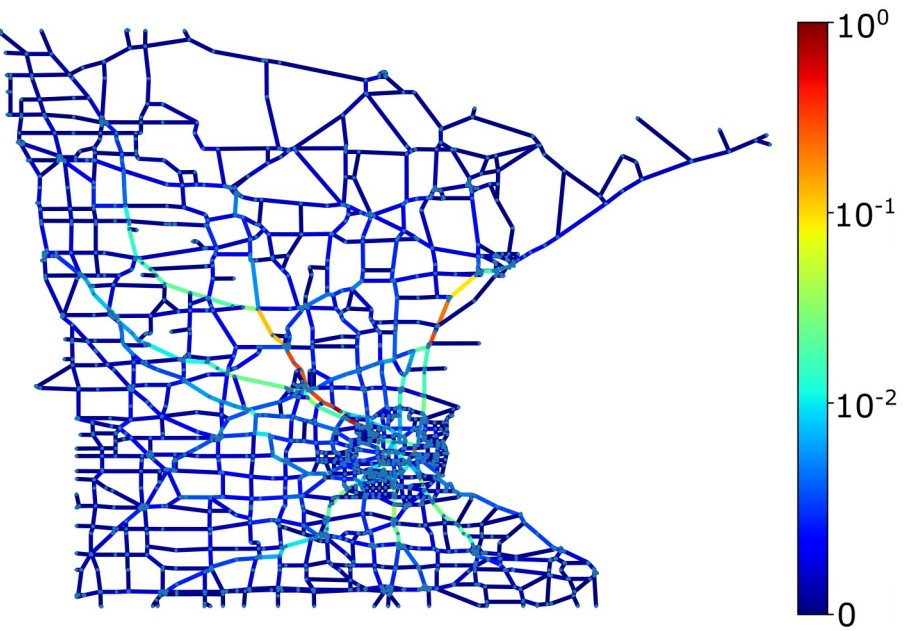

**Fig 16. Edge betweenness centrality score for Minnesota State transportation network.**

In this study, we tested our framework on the Minnesota state network, depicted in Fig 16, using the previously described scenarios Case-I and Case-II for both undirected and bidirected graphs. Fig 16 shows the EBC scores for the graph, and the evaluation metric is summarized in Table 6 and Fig 17. As with the previous case, the GNN method yields high correlation values, showing that the GNN method performs well even for such a large graph. However, importantly, the vulnerability-based metrics cannot be calculated for this graph even using the computational resources deployed for this exercise. This underscores the advantage of the GNN method to evaluate large networks quickly, which may be necessary for emergency response or now-casting for natural disasters.

The Spearman's coefficient of our GNN-based framework on additional four worldwide networks are shown in Table 7. The average Spearman correlation exceeds 0.9, which underscores the performance of our approach.

## Applications to resilience and post-disaster recovery

Rapid identification of critical road segments is crucial for emergency response management to allow timely mitigating actions such as adjusting traffic signal timings or re-routing traffic [92]. This section describes two concrete applications of the proposed method: inter-disaster road resiliency enhancement and post-disaster recovery. As shown in the ablation studies, the GNN-based edge ranking can be applied to calculate EBC for both large and small networks

**Table 6. Spearman's coefficient on minnesota transportation network.**

| Graph | Case I | | Case II | |
|---|---|---|---|---|
| | **Undirected** | **Bidirected** | **Undirected** | **Bidirected** |
| Train | 0.91 ± 0.008 | 0.88 ± 0.011 | 0.88 ± 0.015 | 0.80 ± 0.019 |
| Test | 0.87 ± 0.013 | 0.83 ± 0.020 | 0.81 ± 0.023 | 0.74 ± 0.029 |

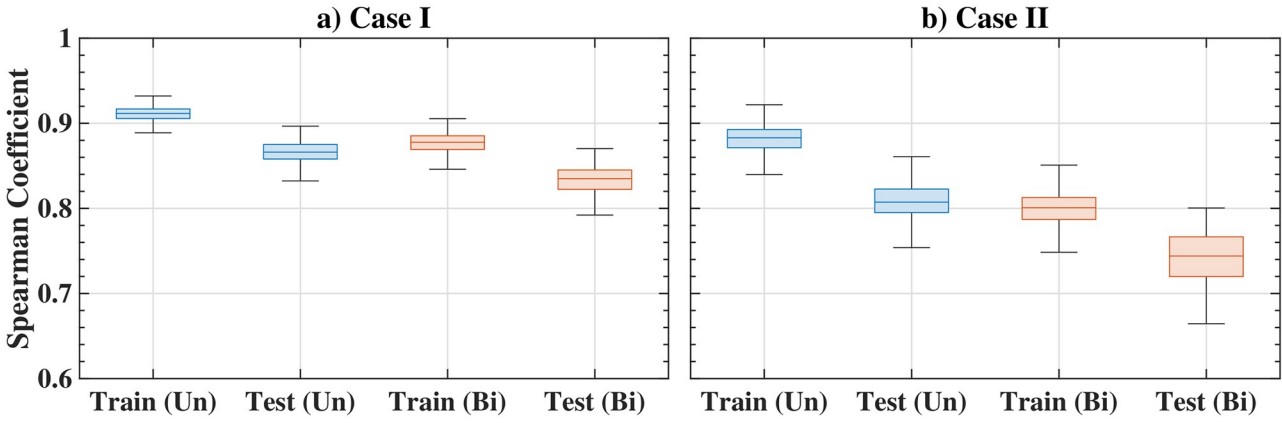

**Fig 17. Edge ranking score distributions for Minnesota road network; also shown are median, the lower and upper quartiles, any outliers (calculated using the interquartile range), and the minimum and maximum values that are not outliers.**

rapidly. This allows it to be applied to understand the effects and mitigate high-frequency traffic flow disruptions such as congestion or for expeditious decision-making during significant events.

In everyday situations and planning without the need for urgency in decision-making, the conventional Brandes' EBC algorithm is sufficient to analyze the graph. However, decision-making under time constraints requires an algorithm to analyze a graph (pre-trained) rapidly. One such example is illustrated using a small graph in Fig 18. For the pre-disaster scenario $G^*$, edges $e_8$, $e_9$, and $e_6$ are ranked 1, 2, and 8, respectively. Following a disruption to $e_8$ rendering it non-operative, $e_9$ is no longer the most critical connection; the edge $e_6$ assumes a new importance rank in the modified graph, and new resource allocation to increase the resiliency of the network is crucial. Otherwise, the risk that the network will be disconnected increases substantially.

Post-disaster recovery sequencing is another application where the proposed GNN method will prove beneficial. This task involves the identification of street segments during every stage of a sequential recovery process, which can be extremely cumbersome to calculate for large networks using conventional methods. To illustrate this idea, we consider the uninterrupted graph $G^*$ shown in Fig 18. The modified (post-disaster) network topology $G$ is shown in Fig 19 (A), where $e_8$, $e_{10}$, and $e_6$ are rendered inoperative. Considering the disrupted edge set as $\mathcal{M}$, where the edges are interchangeably represented as elements of set $\mathcal{M}$ as, $e_8 \leftrightarrow m_1$, $e_{10} \leftrightarrow m_2$,

**Table 7. Spearman's coefficient on Aachen, Edinburgh, Luxembourg, and Santa Barbara Transportation networks.**

| City | Graph | Case I | | Case II | |
|---|---|---|---|---|---|
| | | **Undirected** | **Bidirected** | **Undirected** | **Bidirected** |
| Aachen | Train | 0.92 ± 0.007 | 0.92 ± 0.007 | 0.90 ± 0.011 | 0.91 ± 0.008 |
| | Test | 0.89 ± 0.021 | 0.89 ± 0.017 | 0.90 ± 0.010 | 0.90 ± 0.010 |
| Edinburgh | Train | 0.94 ± 0.005 | 0.95 ± 0.005 | 0.92 ± 0.009 | 0.92 ± 0.010 |
| | Test | 0.91 ± 0.012 | 0.92 ± 0.011 | 0.92 ± 0.010 | 0.92 ± 0.012 |
| Luxembourg | Train | 0.88 ± 0.008 | 0.89 ± 0.008 | 0.87 ± 0.009 | 0.87 ± 0.010 |
| | Test | 0.85 ± 0.014 | 0.85 ± 0.019 | 0.86 ± 0.011 | 0.86 ± 0.011 |
| Santa Barbara | Train | 0.93 ± 0.006 | 0.95 ± 0.004 | 0.92 ± 0.009 | 0.93 ± 0.009 |
| | Test | 0.92 ± 0.009 | 0.93 ± 0.007 | 0.92 ± 0.016 | 0.93 ± 0.011 |

## (A) Pre-disaster network topology (G*)  (B) Inter-disaster network topology (G)

## Network edge connections and weights

| Edge | $e_1$ | $e_2$ | $e_3$ | $e_4$ | $e_5$ | $e_6$ | $e_7$ | $e_8$ | $e_9$ | $e_{10}$ | $e_{11}$ | $e_{12}$ | $e_{13}$ |
|---|---|---|---|---|---|---|---|---|---|---|---|---|---|
| End Nodes | (a,b) | (a,c) | (a,d) | (b,c) | (b,d) | (b,f) | (c,d) | (c,e) | (e,f) | (e,h) | (f,g) | (f,h) | (g,h) |
| Weight | 0.6 | 0.4 | 0.4 | 0.2 | 0.3 | 0.9 | 0.1 | 0.6 | 0.4 | 0.3 | 0.7 | 0.3 | 0.9 |

## Edge Importance Rank of pre-disaster Network (G*)

| Rank | 1 | 2 | 3 | 4 | 5 | 6 | 7 | 8 | 9 | 10 | 11 | 12 | 13 |
|---|---|---|---|---|---|---|---|---|---|---|---|---|---|
| Edge | $e_8$ | $e_9$ | $e_{11}$ | $e_{10}$ | $e_7$ | $e_3$ | $e_4$ | $e_6$ | $e_{13}$ | $e_{12}$ | $e_5$ | $e_2$ | $e_1$ |

## Edge Importance Rank of inter-disaster Network (G) – after $e_8$ is interrupted

| Rank | 1 | 2 | 3 | 4 | 5 | 6 | 7 | 8 | 9 | 10 | 11 | 12 |
|---|---|---|---|---|---|---|---|---|---|---|---|---|
| Edge | $e_6$ | $e_4$ | $e_{11}$ | $e_9$ | $e_{12}$ | $e_5$ | $e_1$ | $e_3$ | $e_7$ | $e_{13}$ | $e_{10}$ | $e_2$ |

**Fig 18. Example scenario for inter-disaster road resiliency improvement; the edge importance ranking order significantly changes with only a minor change in the graph topology, underscoring the need to rapidly estimate edge importance ranking.**

and $e_6 \leftrightarrow m_3$, the importance rank for the three edges are 1, 4, and 8, respectively. Recovery sequencing attempts to answer the cardinal question of the order in which to restore functionality to the disrupted segments. This process should also guide the actions in the event one or more segments identified cannot be recovered.

**Algorithm 2 Sequential repairment procedure for post-disaster recovery**

**Input:** Full edge set $\mathcal{N}$ with cardinality (number of edges) $|\mathcal{N}| = N$ of the original network $G^*$ and associated edge connection list and edge weight list, disrupted edge set $\mathcal{M}$ with cardinality (number of disrupted edges) $|\mathcal{M}| = M$. $\mathcal{M} \subseteq \mathcal{N}$

**Output:** Rank of edges for recovery in every repair stage.

1: Network of $N$ edges has $M$ damaged links, hence the post-disaster network $G$ has $(N - M)$ links. For the first repairment stage, there are $M$ possibilities.

2: Create $M$ different graphs with only one recovered edge; such graphs have $(N - M + 1)$ links. Graph $G_i$ will have the $(N - M)$ uninterrupted edges and one recovered $m_i$, $m_i \in \mathcal{M}$. Here, $G_i$ denotes the disrupted network with recovered $m_i$ in the disrupted edge set. Also, the $m_i$ is associated with one edge, say $e_x$ in $G^*$, $e_x \in \mathcal{N}$

3. Find the rank $(R_i)$ of the recovered edge $m_i$ (or $e_x$) in the respective graph $G_i$.

```
4: Use a comparison-based sorting algorithm based on ranking (R_i) from
   the previous graph. The sorted list should be the repair priority
   for this stage.
5: If R_i for m_i (or e_x) from G_i is the same as R_j for m_j (or e_y) from G_j,
    then construct a graph G_ij with recovered m_i and m_j. For this new
   graph, R_i and R_j are calculated, and the tie is settled. If R_i = R_j
   in G_ij, then both m_i and m_j are assigned the same rank.
6: For 3-way ties, where R_i from G_i, R_j from G_j, and R_k for m_k (or e_z)
   from G_k are all the same, three graphs are created G_ij, G_jk, G_ik and
   the same procedure as Step 5 is followed to obtain the rank of
   repair for m_i, m_j, and m_k for a specific repair stage. This can be
   extended to p-way ties, where (p 2) graphs are created to settle the
   ties.
7: This procedure yields the ranking order for the link repairment in
   the current stage. It is recommended to repair the link with Rank 1
   in this list. However, if this is not possible, then the link with
   Rank 2 should be repaired to maximize network efficiency.
8: In the next stage, with one recovered edge, we have (M − 1)
   disrupted edges. Repeat this whole process until all the edges are
   repaired.
```

We propose Algorithm 2 to address such crucial questions related to post-disaster recovery sequencing. To better explain the idea, the process is explained in relation to the simple graph described previously. As there are three inoperative edges, three scenarios are created (Fig 19 (B.1)–19(B.3)); in each case, only one of the edges is repaired. The rank of the repaired edges on the partially recovered network determines the importance of repairment. From Cases B.1, B.2, and B.3, it is clear that for the partially recovered network, $m_1$ ($e_8$) and $m_3$ ($e_6$) contribute more compared to $m_2$ ($e_{10}$), respectively for $G_1$, $G_3$, and $G_2$. Hence $m_2$ ($e_{10}$) is the lowest rank from a repairment standpoint. Because $m_1$ ($e_8$) and $m_3$ ($e_6$) are both ranked 1 in $G_1$ and $G_3$, the tie scenario (B.4) also needs to be explored. Here, both links are repaired ($G_{13}$), and their ranks are compared; in this case, $m_1$ ($e_8$) and $m_3$ ($e_6$) are ranked 1 and 7, respectively, indicating $m_1$ ($e_8$) is more important than $m_3$ ($e_6$). Hence, the repairment sequencing is $e_8$, $e_6$, and $e_{10}$. This means $e_8$ should be repaired first; in case $e_8$ cannot be repaired, then $e_6$ should be repaired. Starting with 100% network efficiency for $E(G^*)$, the disrupted network efficiency $E(G)$ becomes 67.72%. For each link repair case, the improved network efficiencies become 93.68%, 88.65%, and 71.33% for $e_8$, $e_6$, and $e_{10}$, respectively. Hence, the rank of first-stage recovery is $e_8$, $e_6$, and $e_{10}$, which matches the recovery rank obtained using GNN. It is important to note that for this small graph and 3-disrupted edges, the rankings were calculated four times. For large networks, clearly, the computational overhead with such ranking calculations and network efficiency estimation will increase significantly and can render conventional ranking methods impractical to apply during emergencies. This further underscores the practical advantages of the GNN method described in this paper.

## Conclusions

Edge importance ranking can be used to improve the overall efficiency of a transportation network in terms of prioritizing the construction and maintenance of roads and bridges, and for identifying potential bottlenecks. In this paper, we propose a novel approach for edge importance ranking in large networks, specifically focusing on streets in a transportation network. We utilize a GNN to learn the complex relationships between different edges and to predict their importance based on the graph connectivity and the shortest path in terms of travel times. This framework is learning-based and can be generalized to different network sizes. Hence, it is robust to dynamic changes in the graph, such as changes in travel times or

**(A) Post-disaster network topology (G)**

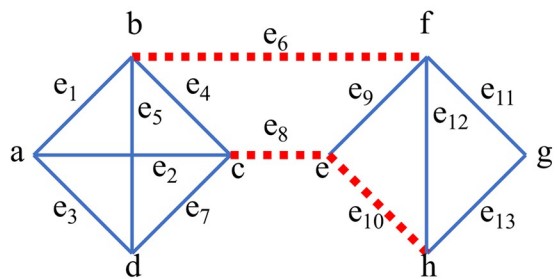

Importance Rank of the damaged edges
in the whole uninterrupted network G*

| Edge | $e_8$ | $e_{10}$ | $e_6$ |
|------|------|------|------|
| Rank | 1 | 4 | 8 |
| Damaged link | $m_1$ | $m_2$ | $m_3$ |

**(B) Repairment Strategy Stage 1 (Ref: Algorithm 2, Step 2-4)**

**(B.1) Scenario-1: $m_1$ ($e_8$) repaired, $m_3$ ($e_6$) and $m_2$ ($e_{10}$) are not repaired ($G_1$)**

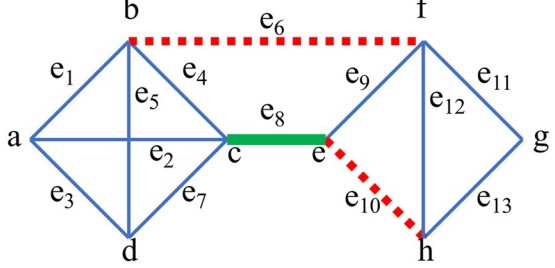

Rank of $m_1$ ($e_8$) in the new network $G_1$ is 1

**(B.2) Scenario-2: $m_2$ ($e_{10}$) repaired, $m_3$ ($e_6$) and $m_1$ ($e_8$) are not repaired ($G_2$)**

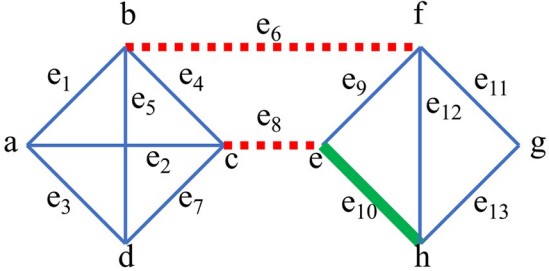

Rank of $m_2$ ($e_{10}$) in the new network $G_2$ is 3

**(B.3) Scenario-3: $m_3$ ($e_6$) repaired, $m_1$ ($e_8$) and $m_2$ ($e_{10}$) are not repaired ($G_3$)**

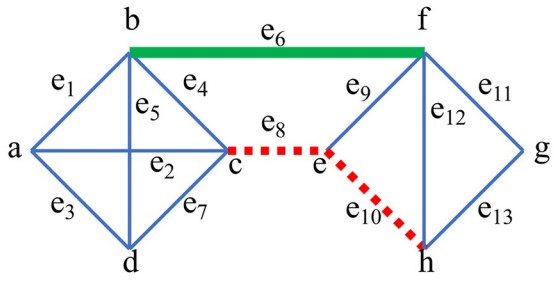

Rank of $m_3$ ($e_6$) in the new network $G_3$ is 1

**(B.4) Tie Scenario (Ref: Algorithm 2, Step 5) : $m_1$ ($e_8$) and $m_3$ ($e_6$) are repaired, $m_2$ ($e_{10}$) is not repaired ($G_{13}$)**

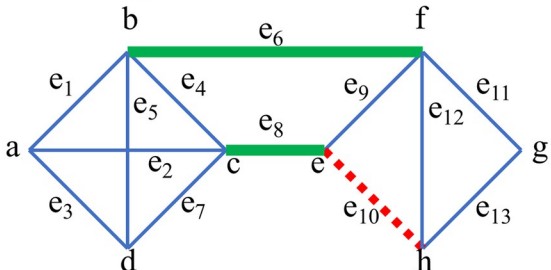

Rank of $m_1$ ($e_8$) and $m_3$ ($e_6$) in the new network $G_{13}$ is 1 and 7, respectively

**Fig 19. Sample application of edge ranking framework in Post-disaster recovery.** The important edges must be identified for every stage of the sequential recovery process multiple times.

disruptions. We evaluated the results from our method in terms of edge ranking metrics such as EBC, network efficiency-based vulnerability, and vulnerability based on the probabilistic measure of distance between networks. Based on several ablation and application experiments, the proposed GNN-based edge importance ranking approach is shown to be

computationally efficient at the expense of only a moderate reduction in accuracy, which yields significant advantages, especially for large graphs. We also present concrete examples to firmly root the ideas to transportation applications to maintenance and recovery sequencing after disasters, where rapid ranking estimation is necessary. Furthermore, this technique could be utilized as a surrogate model for resilience and vulnerability studies involving uncertainty, such as earthquakes. This highlights its versatility and potential for broader use in assessing and enhancing transportation network resilience. As a direction for future exploration, we plan to evaluate and potentially integrate the GNN-based EBC with other traffic-related information like population distribution. Additionally as a scope of future work, we intend to also integrate origin-destination demand data and equity-related information to compare our GNN based approach with egalitarian methods. Based on our study, we can conclude that the GNN approach shows tremendous promise in networked transportation infrastructure applications where time is of the essence, including for resiliency assessments and post-disaster recovery sequencing.

## Appendix: Ablation studies on synthetic graphs

We performed a suite of experiments on synthetic graphs to study the effect of hyper-parameters on the GNN model's performance. The main hyper-parameters of the model are the number of GNN layers and the number of embedding dimensions; we vary these hyper-parameters and evaluate the performance of the model. We use Erdős—Rényi variant-I (GNP random) graphs for this study.

### Varying number of layers

The number of GNN layers in the model influences the amount of information any given edge can accumulate from its neighboring edges. For an increasing number of GNN layers, the edges have access to information from multi-hop adjacent edges. We varied the number of GNN layers from 1 to 5, while keeping the embedding dimension fixed (256). Additionally, we present the model performance in Fig 20(a). Both evaluation metrics, i.e., Kendall tau and Spearman's correlation coefficient, show that models with small numbers of GNN layers perform poorly, as the feature aggregation reach for each edge is limited. Increasing the number of GNN layers yields better ranking performance. Therefore, we fix the number of GNN layers as 5 for all the numerical and experimental studies; increasing this number further comes at the cost of higher training time with only a marginal improvement in accuracy.

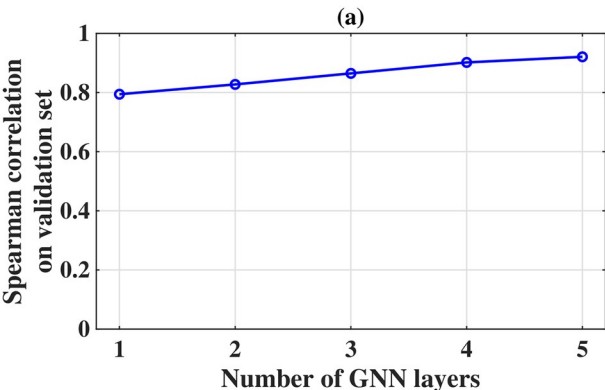 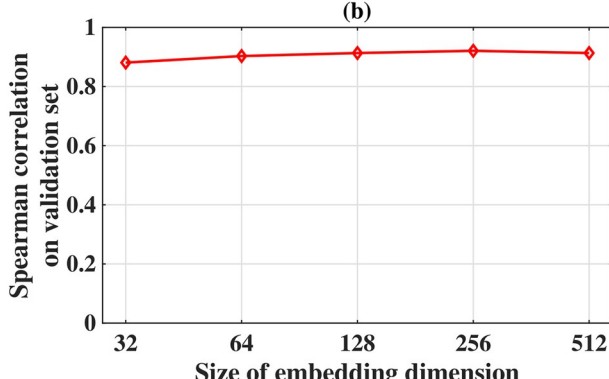

**Fig 20. Evaluation metric for different number of GNN layers.**

## Varying embedding dimensions

For any neural network, the embedding dimension (number of neurons in the hidden layers) represents the number of learnable modal parameters. Under-parameterized models (low embedding dimension) cannot approximate complex functions, whereas over-parameterized models (high embedding dimension) generalize poorly. In this experiment, we change the embedding dimension to 32, 64, 128, 256, and 512, with five layers (obtained from). We evaluate the performance for all these trials and present it in Fig 20(b). Results show that an embedding dimension of 256 is optimal for our case, with degrading performance for both the lower and higher embedding dimensions.

## Author Contributions

**Conceptualization:** Debasish Jana, Sven Malama, Sriram Narasimhan, Ertugrul Taciroglu.

**Data curation:** Debasish Jana, Sven Malama.

**Formal analysis:** Debasish Jana, Sven Malama.

**Funding acquisition:** Sriram Narasimhan, Ertugrul Taciroglu.

**Investigation:** Debasish Jana, Sven Malama, Sriram Narasimhan, Ertugrul Taciroglu.

**Methodology:** Debasish Jana, Sven Malama, Sriram Narasimhan.

**Project administration:** Sriram Narasimhan, Ertugrul Taciroglu.

**Resources:** Sriram Narasimhan, Ertugrul Taciroglu.

**Software:** Debasish Jana, Sven Malama.

**Supervision:** Sriram Narasimhan, Ertugrul Taciroglu.

**Validation:** Debasish Jana, Sven Malama.

**Visualization:** Debasish Jana, Sven Malama.

**Writing – original draft:** Debasish Jana.

**Writing – review & editing:** Sven Malama, Sriram Narasimhan, Ertugrul Taciroglu.

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
