## [Decision Letter · Decision Letter 0]

1 Sep 2023

PONE-D-23-21688Edge-based graph neural network for ranking critical road segments in a networkPLOS ONE

Dear Dr. Jana,

Thank you for submitting your manuscript to PLOS ONE. After careful consideration, we feel that it has merit but does not fully meet PLOS ONE’s publication criteria as it currently stands. Therefore, we invite you to submit a revised version of the manuscript that addresses the points raised during the review process.

We look forward to receiving your revised manuscript.

Kind regards,

Wei Ju, Ph.D.

Academic Editor

PLOS ONE

“We gratefully acknowledge the funding support provided by the City of Los Angeles’s

Bureau of Engineering (BOE).”

6. We note that Figures 13 and 15 in your submission contain [map/satellite] images which may be copyrighted. All PLOS content is published under the Creative Commons Attribution License (CC BY 4.0), which means that the manuscript, images, and Supporting Information files will be freely available online, and any third party is permitted to access, download, copy, distribute, and use these materials in any way, even commercially, with proper attribution. For these reasons, we cannot publish previously copyrighted maps or satellite images created using proprietary data, such as Google software (Google Maps, Street View, and Earth). For more information, see our copyright guidelines: http://journals.plos.org/plosone/s/licenses-and-copyright.

a. You may seek permission from the original copyright holder of Figures 13 and 15 to publish the content specifically under the CC BY 4.0 license. 

Reviewers' comments:

Reviewer's Responses to Questions

**Comments to the Author**

1. Is the manuscript technically sound, and do the data support the conclusions?

Reviewer #1: Yes

Reviewer #2: Yes

2. Has the statistical analysis been performed appropriately and rigorously? 

Reviewer #1: Yes

Reviewer #2: Yes

3. Have the authors made all data underlying the findings in their manuscript fully available?

Reviewer #1: Yes

Reviewer #2: Yes

4. Is the manuscript presented in an intelligible fashion and written in standard English?

Reviewer #1: Yes

Reviewer #2: Yes

5. Review Comments to the Author

Reviewer #1: This paper studies the task of ranking critical road segments for transportation networks. The authors pose the transportation network as a graph with roads as edges and intersections as nodes and deploy a Graph Neural Network (GNN) for the task.

Strength:

1. Ranking critical road segments is indeed a challenging and meaningful task for transportation networks. GNNs are suitable approaches for the task.

2. The idea of the paper is reasonable and easy to follow.

3. The paper presents the result on both synthetic and real-world graph data, experimental settings (i.e., hyper-parameters and complexity analysis) are provided.

4. Some applications of the proposed framework are presented.

Questions:

1. The paper generates 1000 training graphs, 100 516 validation graphs, and 100 test graphs. What's the difference with a single real-world road network (i.e., U.S. Freeway network), what are the train, valid and test dataset details of the real-world network?

Weakness:

1. It is better to present the detail of generated synthetic and real-world graph data in two tables.

2. More experiments are needed like representation visualization to prove the effectiveness of the proposed framework.

3. It is highly recommended for authors include more reviews about GNNs in related work [1-7] and compare more SOTA baselines to show the effectiveness of the provided method.

[1] Dynamic Hypergraph Structure Learning for Traffic Flow Forecasting. ICDE 2023

[2] Analysis of robustness of urban bus network. Chinese Physics B 2016

[3] A Comprehensive Survey on Deep Graph Representation Learning. arXiv 2023

[4] Kernel-based Substructure Exploration for Next POI Recommendation. ICDM 2022

[5] DisenHAN: Disentangled Heterogeneous Graph Attention Network for Recommendation. CIKM 2020

[6] DisenCite: Graph-based Disentangled Representation Learning for Context-Specific Citation Generation. AAAI 2022

[7] DisenCTR: Dynamic Graph-based Disentangled Representation for Click-Through Rate Prediction. SIGIR 2022

[8] HE-SNE: Heterogeneous Event Sequence-based Streaming Network Embedding for Dynamic Behaviors. IJCNN 2022

Reviewer #2: This paper focuses on a interesting topic, which is identifying critical and vital road segments in a

transportation network. The problem is solved via a transportation graph with roads as edges and intersections as nodes and deploy a GNN to rank the importance of road segments. The proposed model is evaluated on synthetic graphs and two real-world transportation networks. However, there exist certain areas in the paper that could be improved.

Firstly, the proposed method is evaluated on three synthetic but only two real-world datasets. Experiments on other real-world dataset are needed for more persuisive results. Additionally, this paper lacks a detailed visualization of the transportation netwrok and ranked segments to intuitively show the effect of applied method.

6. PLOS authors have the option to publish the peer review history of their article (what does this mean?). If published, this will include your full peer review and any attached files.

Reviewer #1: No

Reviewer #2: No

---

## [Author Response · Author response to Decision Letter 0]

2 Oct 2023

We thank the academic editor and the reviewers for handling and reviewing the manuscript. We have provided our response to reviewers' questions and comments in the "Response to Reviewers" document.

---

## [Decision Letter · Decision Letter 1]

29 Nov 2023

PONE-D-23-21688R1Edge-based graph neural network for ranking critical road segments in a networkPLOS ONE

Dear Dr. Jana,

Thank you for submitting your manuscript to PLOS ONE. After careful consideration, we feel that it has merit but does not fully meet PLOS ONE’s publication criteria as it currently stands. Therefore, we invite you to submit a revised version of the manuscript that addresses the points raised during the review process.

We look forward to receiving your revised manuscript.

Kind regards,

Praveen Kumar Donta, Ph.D.

Academic Editor

PLOS ONE

Journal Requirements:

Reviewers' comments:

Reviewer's Responses to Questions

**Comments to the Author**

1. If the authors have adequately addressed your comments raised in a previous round of review and you feel that this manuscript is now acceptable for publication, you may indicate that here to bypass the “Comments to the Author” section, enter your conflict of interest statement in the “Confidential to Editor” section, and submit your "Accept" recommendation.

Reviewer #1: All comments have been addressed

Reviewer #3: All comments have been addressed

Reviewer #4: All comments have been addressed

2. Is the manuscript technically sound, and do the data support the conclusions?

Reviewer #1: Yes

Reviewer #3: Yes

Reviewer #4: Yes

3. Has the statistical analysis been performed appropriately and rigorously? 

Reviewer #1: Yes

Reviewer #3: Yes

Reviewer #4: Yes

4. Have the authors made all data underlying the findings in their manuscript fully available?

Reviewer #1: Yes

Reviewer #3: Yes

Reviewer #4: Yes

5. Is the manuscript presented in an intelligible fashion and written in standard English?

Reviewer #1: Yes

Reviewer #3: Yes

Reviewer #4: Yes

6. Review Comments to the Author

Reviewer #1: (No Response)

Reviewer #3: I think the authors have addressed the reviewers' concerns and this version can be accepted for publication at PONE.

Reviewer #4: The authors proposed a GNN-based system to calculate the edge betweenness certainty in transportation networks. They have explained the methodology in detail, accompanied by proper diagrams and algorithms, in a clear and comprehensive manner. Additionally, the authors conducted evaluations on both synthetic and real-world data, presenting the results in detail. However, several issues have been identified in the manuscript, as follows:

1. The authors are advised to include a paragraph about road network vulnerability analysis and its importance. Although, the introduction paragraph of the manuscript covers certain aspects insufficiently, making it challenging for a general reader to grasp the significance of this work.

2. The proposed GNN model is trained for only 50 epochs. However, according to Fig. 10(a), the training loss is steeply decreasing with each epoch. Therefore, if the model is trained for more epochs, it can optimize well and perform better. The authors are advised to train their model for more epochs and assess its performance.

3. In the training phase of the proposed approach, the desired output was calculated by utilizing the conventional method for determining edge betweenness certainty (EBC). However, in Fig. 16, a comparison is made between conventional EBC-based ranking and the ranking obtained through the GNN. The authors need to justify this contradiction in detail.

7. PLOS authors have the option to publish the peer review history of their article (what does this mean?). If published, this will include your full peer review and any attached files.

Reviewer #1: No

Reviewer #3: **Yes: **Cancheng Li

Reviewer #4: No

---

## [Author Response · Author response to Decision Letter 1]

1 Dec 2023

We thank the editors and the reviewers for handling and reviewing the manuscript for technical merits. We have responded to all the questions and comments raised during the technical round.

---

## [Decision Letter · Decision Letter 2]

5 Dec 2023

Edge-based graph neural network for ranking critical road segments in a network

PONE-D-23-21688R2

Dear Dr. Jana,

We’re pleased to inform you that your manuscript has been judged scientifically suitable for publication and will be formally accepted for publication once it meets all outstanding technical requirements.

Kind regards,

Praveen Kumar Donta, Ph.D.

Academic Editor

PLOS ONE

Additional Editor Comments (optional):

Reviewers' comments:

Reviewer's Responses to Questions

**Comments to the Author**

1. If the authors have adequately addressed your comments raised in a previous round of review and you feel that this manuscript is now acceptable for publication, you may indicate that here to bypass the “Comments to the Author” section, enter your conflict of interest statement in the “Confidential to Editor” section, and submit your "Accept" recommendation.

Reviewer #4: All comments have been addressed

2. Is the manuscript technically sound, and do the data support the conclusions?

Reviewer #4: Yes

3. Has the statistical analysis been performed appropriately and rigorously? 

Reviewer #4: Yes

4. Have the authors made all data underlying the findings in their manuscript fully available?

Reviewer #4: Yes

5. Is the manuscript presented in an intelligible fashion and written in standard English?

Reviewer #4: Yes

6. Review Comments to the Author

Reviewer #4: All of my concerns are clearly addressed and well-written. If possible, please improve the resolution of Fig. 15.

7. PLOS authors have the option to publish the peer review history of their article (what does this mean?). If published, this will include your full peer review and any attached files.

Reviewer #4: No

---

## [Editor Report · Acceptance letter]

12 Dec 2023

PONE-D-23-21688R2 

Edge-based graph neural network for ranking critical road segments in a network 

Dear Dr. Jana:

I'm pleased to inform you that your manuscript has been deemed suitable for publication in PLOS ONE. Congratulations! Your manuscript is now with our production department. 

Kind regards, 

on behalf of

Dr. Praveen Kumar Donta 

Academic Editor

PLOS ONE